



# A global catalogue of large SO₂ sources and emissions derived from the Ozone Monitoring Instrument

Vitali E. Fioletov[1], Chris A. McLinden[1], Nickolay Krotkov[2], Can Li[2,3], Joanna Joiner[1], Nicolas Theys[4], Simon Carn[5,6], and Mike D. Moran[1]

[1]Air Quality Research Division, Environment Canada, Toronto, ON, Canada
[2]Atmospheric Chemistry and Dynamics Laboratory, NASA Goddard Space Flight Center, Green-belt, MD, USA
[3]Earth System Science Interdisciplinary Center, University of Maryland, College Park, MD, USA
[4]Belgian Institute for Space Aeronomy (BIRA-IASB), Brussels, Belgium
[5]Department of Geological and Mining Engineering and Sciences, Michigan Technological University, Houghton, MI 49931, USA
[6]Department of Mineral Sciences, National Museum of Natural History, Smithsonian Institution, Washington, DC 20560, USA

*Correspondence to*: Vitali Fioletov (Vitali.Fioletov@outlook.com or Vitali.Fioletov@canada.ca)

**Abstract.** Sulphur dioxide (SO₂) measurements from the Ozone Monitoring Instrument (OMI) satellite sensor processed with the new Principal Component Analysis (PCA) algorithm were used to detect large point emission sources or clusters of sources. The total of 491 continuously emitting point sources releasing from about 30 kt yr⁻¹ to more than 4000 kt yr⁻¹ of SO₂ per year have been identified and grouped by country and by primary source origin: volcanoes (76 sources); power plants (297); smelters (53); and sources related to the oil and gas industry (65). The sources were identified using different methods, including through OMI measurements themselves applied to a new emissions detection algorithm, and their evolution during the 2005-2014 period was traced by estimating annual emissions from each source. For volcanic sources, the study focused on continuous degassing, and emissions from explosive eruptions were excluded. Emissions from degassing volcanic sources were measured, many for the first time, and collectively they account for about 30% of total SO₂ emissions estimated from OMI measurements, but that fraction has increased in recent years given that cumulative global emissions from power plants and smelters are declining while emissions from oil and gas industry remained nearly constant. Anthropogenic emissions from the USA declined by 80% over the 2005-2014 period as did emissions from western and central Europe, whereas emissions from India nearly doubled, and emissions from other large SO₂-emitting regions (South Africa, Russia, Mexico, and the Middle East) remained fairly constant. In total, OMI-based estimates account for about a half of total reported anthropogenic SO₂ emissions; the remaining half is likely related to sources emitting less than 30 kt yr⁻¹ and not detected by OMI.



# 1. Introduction

The concept of monitoring Sulphur dioxide (SO$_2$) and other gaseous pollutants from satellite using remote sensing in UV and IR spectral bands was suggested long before satellite instruments capable of such measurements were launched (Barringer and Davies, 1977). The first satellite measurements of SO$_2$ were reported in 1979, although these measurements were by

the Voyager 1 satellite of the atmosphere of Jupiter's moon Io (Bertaux and Belton, 1979). In the Earth's atmosphere, the El Chichon volcanic eruption in 1983 injected a large amount of SO$_2$ into the atmosphere that was detected by the Total Ozone Mapping Spectrometer (TOMS) (Krueger, 1983) and the Solar Backscattered Ultraviolet (SBUV) instrument (McPeters et al., 1984), both on board NASA's Nimbus 7 satellite. In the following years, data from TOMS on board Nimbus 7 and Earth Probe satellites were used to monitor SO$_2$ emissions from explosive and non-explosive volcanic eruptions (Bluth and Carn,

2008; Bluth et al., 1992, 1993; Carn et al., 2003; Krueger et al., 1995; Yang et al., 2007). It was also shown that TOMS could detect anthropogenic SO$_2$ emissions, although only when atmospheric loadings were exceptional (Carn et al., 2004).

Global Ozone Monitoring Experiment (GOME) measurements on the Earth Research Satellite 2 (ERS-2), which started in 1995, demonstrated that anthropogenic SO$_2$ sources such as power plants in Eastern Europe (Eisinger and Burrows, 1998) and smelters in Peru and Russia (Khokhar et al., 2008) can be monitored from space. The past 15 years have seen the

launch of three satellite UV instruments capable of detecting near-surface SO$_2$: the SCanning Imaging Ab-sorption spectroMeter for Atmospheric CHartographY (SCIAMACHY), 2002-2012, on board the ENVISAT satellite (Bovensmann et al., 1999); the Global Ozone Monitoring Experiment-2 (GOME 2) instrument, 2006-present, on MetOp-A (Callies et al., 2000); and the Ozone Monitoring Instrument (OMI) (Levelt et al., 2006), 2004-present, on NASA's Aura spacecraft (Schoeberl et al., 2006). OMI provides daily, nearly global maps of vertical column densities of SO$_2$, has the highest spatial

resolution, longest operation, and lowest degradation, and is the most sensitive to SO$_2$ sources among the satellite instruments of its class (Fioletov et al., 2013).

Vertical column densities of SO$_2$ can be also retrieved from satellite measurements in the thermal infrared (IR) parts of the spectrum. This type of measurement, utilized for example by the Infrared Atmospheric Sounding Interferometer (IASI) instrument, was used to trace SO$_2$ from volcanic eruptions (Clarisse et al., 2012; Karagulian et al., 2010),

transcontinental transport of SO$_2$ pollution from China (Clarisse et al., 2011), and anthropogenic emissions from Norilsk, Russia (Bauduin et al., 2014). However, as measurements in the IR are based on the temperature contrast between the surface and air above, they have reduced sensitivity to the boundary layer and hence are only able to detect the largest of sources.

Satellite measurements of trace gases are increasingly employed to monitor emissions (Streets et al., 2013). In

particular, satellite SO$_2$ observations are widely used to calculate volcanic SO$_2$ budgets and to track plumes from volcanic eruptions (e.g., Carn et al., 2003; Rix et al., 2012). A review of different techniques to derive volcanic SO$_2$ fluxes using satellite measurements of plumes of SO$_2$ and to investigate the temporal evolution of the total emissions of SO$_2$ was presented recently by (Theys et al., 2013). It is more challenging, however, to monitor emissions from relatively small



anthropogenic sources. OMI $SO_2$ data were used to study the evolution of regional emissions for China (Jiang et al., 2012; Li et al., 2010; Witte et al., 2009), India (Lu et al., 2013), and USA (Fioletov et al., 2011). It was also demonstrated that satellite instruments can detect $SO_2$ signals from multiple anthropogenic point sources such as power plants, copper and nickel smelters, Canadian oil sand mines, and other sources (Carn et al., 2004, 2007; Fioletov et al., 2013; de Foy et al., 2009; Lee et al., 2009; McLinden et al., 2012, 2014; Nowlan et al., 2011; Thomas et al., 2005) and estimate emissions from them. In this study, we applied a technique based on an exponentially-modified Gaussian fit (Beirle et al., 2014; de Foy et al., 2014) adapted to monitor relatively small anthropogenic sources (Fioletov et al., 2015; de Foy et al., 2015; Lu et al., 2015).

The Principal Component Analysis (PCA) algorithm recently developed by NASA (Li et al., 2013) has substantially reduced noise and eliminated most of the artefacts seen in the previous OMI $SO_2$ data products. This has made OMI $SO_2$ products even more suitable for monitoring anthropogenic sources that emit as little as 30 kt $yr^{-1}$ (Fioletov et al., 2015). OMI measurements were also recently re-processed by the Belgian Institute for Space Aeronomy (BIRA) with a new Differential Optical Absorption Spectroscopy (DOAS) $SO_2$ algorithm, which is a prototype of the operational $SO_2$ algorithm planned for use with the ESA's Sentinel 5 Precursor (TROPOMI) mission (Theys et al., 2015). While most of the results in this study are based on NASA PCA data, the BIRA DOAS data set was also tested.

Global and regional $SO_2$ and $NO_2$ spatial distributions and their time evolution based on OMI observations are discussed in this issue (Krotkov et al., 2015). Unlike $NO_2$ maps, where sources seen by OMI are numerous, global large-scale $SO_2$ maps are not very informative because high $SO_2$ values are observed only in the vicinity of a relatively small number (compared to $NO_2$) of point sources, while values elsewhere are below the OMI detectability limits (Figure 1). There are several regions such as eastern China and the eastern U.S. that contain hundreds of individual $SO_2$ sources (mostly power plants) as discussed in detail by (Krotkov et al., 2015), but these regions are exceptions.

In this study we have catalogued the major $SO_2$ "hot spots" seen by OMI and attributed them to known point sources of $SO_2$ emissions. A recently developed method for detection of sources based on comparison of upwind and downwind $SO_2$ amounts was also used to find the exact location of sources (McLinden et al., 2016). Several databases of volcanoes, power plants, mining and smelting sites, etc., were used for the attribution. OMI data were also used to estimate emissions from these sources and their evolution in time. The 2005-2007 period was used as the main period to detect the sources, and then time series of annual $SO_2$ emissions were estimated for each source for the 2005-2014 period.

The satellite data sets and ancillary meteorological information used in this study are de-scribed in Section 2, and the emission estimation algorithm is described in Section 3. Section 4 compares the NASA PCA and BIRA DOAS OMI data sets. Section 5 discusses four main types of anthropogenic and natural $SO_2$ sources and provides specific examples of sources for each type. In addition to well-known sources, we selected typical but seldom discussed sources. We also used these examples to illustrate the emission estimation method and practical aspects of its application. The catalogue itself is discussed in Section 6 and is provided in the Supplement to this study, including detailed information on the source location, source type, and estimated annual emissions and their uncertainties for 2005-2014 in the form of an electronic spreadsheet.



The emissions estimates and their comparisons with available inventories are then discussed in Section 7, and Section 8 summarizes the results.

## 2. Data sources

5 OMI $SO_2$ data. The new-generation operational OMI Planetary Boundary Layer (PBL) $SO_2$ data produced with the Principal Component Analysis (PCA) algorithm (Li et al., 2013) for the period 2005-2014 were used in this study. Retrieved $SO_2$ vertical column density (VCD) values are given as total column $SO_2$ in Dobson Units (DU, 1 DU = 2.69•1026 molec•km-2). The standard deviation of PCA-retrieved background $SO_2$ is ~0.5 DU, about half the noise of the previous NASA $SO_2$ data product (Li et al., 2013). Although the PCA algorithm uses spectrally-dependent $SO_2$ Jacobians instead of an air mass factor
10 (AMF) as in the previous operational OMI Band Residual Difference (BRD) algorithm, its current version assumes the same fixed conditions as those in the BRD algorithm to facilitate the comparison between the two algorithms (see (Li et al., 2013), for details). The PCA retrievals can therefore be interpreted as having an effective AMF of 0.36 that is representative of typical summertime conditions in the eastern U.S. (Krotkov et al., 2006). This approach, however, results in systematic errors for sources located at high elevations or different latitudes or having different surface conditions.

15  In addition to the standard NASA PCA data set based on a constant AMF=0.36, for this study we have scaled constant PCA $SO_2$ AMF to a source-specific value using two methods of different complexity. The first is a simple correction to account for the elevation of the source, very often the most critical parameter in the $SO_2$ AMF calculation. The second is a more comprehensive treatment in which other factors such as surface reflectivity, solar zenith angle, viewing geometry, surface pressure, cloud fraction and pressure, and the $SO_2$ profile shape were also ac-counted for. This second
20 method follows the approach in McLinden et al. (2014), except that here the $SO_2$ profile is estimated based on the elevation of the source and the climatological boundary layer height (specific to the source location and the time of day of the source) (von Engeln and Teixeira, 2013). Between these two heights the profile is assumed to have a constant mixing ratio while outside of these heights it is assumed to be zero. This comprehensive treatment was used for the emission estimates presented in this paper. Differences between these different approaches to specifying AMF are discussed in the uncertainty
25 analysis given in section 3. Note that the catalogue data file in the Supplement contains a column with the AMF values calculated using this second approach. Also, site-specific AMF values were calculated for the catalogue sites only, and the regional $SO_2$ maps used for illustrations are based on the original PCA product with AMF=0.36.

  In addition to PCA algorithm, $SO_2$ data were also produced using the BIRA DOAS algorithm developed by the Belgian Institute for Space Aeronomy (BIRA-IASB). The retrieval of $SO_2$ slant columns is made in the wavelength interval
30 312-326 nm and includes spectra for $SO_2$, O3 absorption, and the Ring effect. Other fitting windows are also used for strong volcanic eruptions, but these retrievals are not considered in this study. In a second step, the retrieved $SO_2$ slant columns are corrected for possible large-scale biases using a time-, row- and O3 absorption-dependent background correction scheme.





Then the corrected slant columns are converted into vertical columns by the means of an AMF calculation that accounts for surface reflectance, surface height, clouds, solar zenith angle, observation angles and $SO_2$ profile shape. Details can be found in Theys et al. (2015).

OMI $SO_2$ data are retrieved for 60 cross-track positions (or rows), and the pixel ground size varies depending on the track position from $13 \times 24$ km² at nadir to about $28 \times 150$ km² at the outermost swath angle. Data from the first 10 and last 10 cross-track positions were excluded from the analysis to limit the across-track pixel width to about 40 km. As well, beginning in 2007, some rows were affected by field-of–view blockage and stray light (the so-called "row anomaly": see http://www.knmi.nl/omi/research/product/rowanomaly-background.php) and the affected pixels were also excluded from the analysis. Only clear-sky data, defined as having a cloud radiance fraction (across each pixel) less than 20%, and only measurements taken at solar zenith angles less than 70° were used. Additional information on the OMI PCA $SO_2$ product is available from (Krotkov et al., 2015).

The PCA algorithm presently does not account for the effects of snow albedo on the $SO_2$ Jacobians. Unless it is stated otherwise, measurements with snow on the ground were excluded from the analysis. Snow information was obtained from the Interactive Multisensor Snow and Ice (IMS) Mapping System (Helfrich et al., 2007). Wind speed and direction data are required to apply the source detection and emission estimation technique used in this study. ECMWF (European Centre for Medium-Range Weather Forecasts) reanalysis data (Dee et al., 2011): http://data-portal.ecmwf.int/data/d/interim_full_daily) were merged with OMI observations. Wind profiles are available every 6 hours on a 0.75° horizontal grid and are interpolated in space and time to the location of each OMI pixel centre. U- and V- (west-east and south-north, respectively) wind-speed components were averaged in the vertical to account for the vertical distribution of the $SO_2$ profile. For this study mean wind components were calculated for four 1-km thick layers above sea level (from 0-1 km to 3-4 km), the appropriate layer was selected based on the site elevation, and then the wind data were interpolated spatially and temporally to the location and overpass time of each OMI pixel. For the few sites over 4 km in elevation, wind data for the 3-4 km layer were used.

## 3. Method description

### 3.1 The wind rotation technique and fitting

Level-2 high-quality OMI PCA data, combined with the pixel averaging or oversampling technique (Fioletov et al., 2011, 2013; de Foy et al., 2009; McLinden et al., 2012), were used for the 2005-2007 period to detect $SO_2$ "hotspots" for the global $SO_2$ source catalogue. This catalogue was compiled by examining global level-2 OMI data gridded on a 0.04° by 0.04° latitude-longitude grid and smoothed with a 0.2°-wide window to identify potential locations where $SO_2$ values were above the threshold level of 0.1 DU. As the standard error of PCA OMI values is about 0.5 DU, a threshold level of 0.1 DU is just above the 3-σ significance level for the 3-year-long period of observations (assuming about 100 days with suitable observational conditions). Roughly 500 locations of elevated $SO_2$ VCD were identified for further study by analysing over-





passes within a 300-km radius of each "hotspot". The overpass data were also used to construct illustrative maps of the mean SO$_2$ distribution in the vicinity of the sources. For these maps, the pixel averaging technique was applied: a grid with 10-km horizontal spacing was established for the 220 km by 220 km area around the source and for each grid point, all pixels cantered within a 25-km radius were averaged and then shown on the map.

The grid was also used to apply the source detection algorithm to search for SO$_2$ sources (McLinden et al., 2016). Each point on the global grid is evaluated as a potential source location by applying the wind rotation technique and then comparing upwind and downwind SO$_2$ values. Then the wind rotation technique was used to verify the sources and to estimate emissions from them. The approach adopted here involves the rotation of each OMI pixel around the source so that after rotation, all have a common wind direction (Fioletov et al., 2015; Pommier et al., 2013; Valin et al., 2013). To apply

this rotation, the wind speed and direction were determined for each satellite pixel. Then all individual OMI pixels were rotated around the source in a way that the wind direction was always from one direction (from the North in our study). The wind speed and direction are not correlated with the SO$_2$ "signal" for spurious sources, whereas SO$_2$ values upwind from a real source should be lower than these downwind from the source. For illustrative purposes, the same pixel averaging method described at the beginning of this section was used to produce the maps after the wind rotation procedure.

With the rotation technique applied, we can analyse the data assuming that the wind al-ways has the same direction in order to estimate the emissions. In this next step, emissions and lifetimes for each of the detected point sources were estimated using the Exponentially-Modified Gaussian fit (Beirle et al., 2013; Fioletov et al., 2015; de Foy et al., 2015). Inferring the emission strength (E) requires knowledge of the total SO$_2$ mass ($a$) near the source and its lifetime or, more accurately, decay time ($\tau$). Assuming a steady state these quantities are related through the equation $E=a/\tau$. The method

used here was based on fitting OMI-measured SO$_2$ vertical column densities to a three-dimensional parameterization function of the horizontal coordinates and wind speed as described by (Fioletov et al., 2015). A Gaussian function $f(x, y)$ multiplied by an exponentially modified Gaussian function $g(y, s)$ was used to fit the OMI SO$_2$ measurements: $OMI_{SO_2} = a \cdot f(x, y) \cdot g(y, s)$, where $x$ and $y$ (in km) refer to the coordinates of the OMI pixel centre across and along the wind direction, respectively, after the rotation along the wind direction was applied and s (in km per hour) is the wind speed at the

pixel centre. Three parameters, $a$, $\lambda = 1/\tau$, and $\sigma$, were estimated from the fit of the observed OMI values by the function OMI$_{SO2}$. While $\tau$ does not represent chemical lifetime and is affected by deposition, advection, and dispersion of the plume, pixel size, etc., it has been demonstrated that this approach can produce accurate estimates of emissions (de Foy et al., 2014). The third parameter $\sigma$ describes the width or spread of the plume.

    The above method for estimating emissions is designed for point sources. However, multiple sources located

within close proximity could yield unrealistic values of $\tau$ and $\sigma$, and a secondary source located downwind from the primary one could lead to an increase in the value of $\tau$. For example, for the Palabora smelter (23.99°S, 31.16°E), South Africa, the decay time is greatly over-estimated (193 hours) due to the influence of a cluster of power plants about 250 km away near Johannesburg. Similarly, multiple sources located within 20-30 km of the primary source may lead to increases in the value



of σ. Figure 2 shows the distribution of the estimated parameters based on the fit of 2005-2007 OMI data for 215 catalogue sites that produced estimates of σ and τ with small uncertainties. The mean value of τ is about 6 hours, and 80% of all values are between 3 and 9.5 hours. Similarly, the mean value of σ is about 20 km, while the 10$^{th}$ and 90$^{th}$ percentiles are 12 km and 31 km, respectively.

5 If we prescribe fixed values of τ and σ, then the only unknown parameter remaining is the total SO$_2$ mass (a) and the fitting task turns into a simple linear regression as OMI$_{SO2}$ depends on a linearly. We then get a very robust algorithm that can be used to estimate annual and even season-al emissions for detectable sources. Moreover, as it is only the total mass that is estimated, the method can be applied to sites with multiple sources. Essentially the algorithm turns into a weighted average of all individual OMI measurements, where the weights are determined by the pixel position and wind

10 speed and direction. The disadvantage of this approach is that we may introduce a systematic error (a scaling factor) if the actual values of τ and σ for a source are different from the prescribed ones. We used prescribed values of τ=6 hours and σ=20 km for the emission estimates.

To estimate errors related to the uncertainty of the τ and σ values, we recalculated emissions for all sources using values that correspond to their 10th and 90th percentiles. A change of τ=6 hours to its 10th-percentile value (3 hours)

15 increases the emission estimate on average by about 50%, while setting τ to its 90th-percentile value (9.5 hours) decreases the estimates by about 25%. Similarly, setting σ to its 10th-percentile value (12 km) and 90th-percentile value (30 km) changes emission estimates by -30% and +30%, respectively.

The parameter estimation was done using OMI pixels cantered within a rectangular area that spreads ±$L$ km across the wind direction, $L$ km in the upwind direction and 3·$L$ km in the downwind direction, and for wind speeds between 0.5

20 and 45 km h$^{-1}$. For better separation of different sources where multiple sources are located in the same area, different values of $L$ were used depending on the emission strength. The value of $L$ was chosen to be 30 km for small sources (under 100 kt yr$^{-1}$), 50 km for medium sources (between 100 kt yr$^{-1}$ and 1000 kt yr$^{-1}$), and 90 km for large sources (more than 1000 kt yr$^{-1}$). For small sources, different $L$ values have little effect on the estimated parameters. For larger sources, pixels with elevated SO$_2$ values are located over larger areas and therefore the parameters estimated for higher $L$ values have smaller uncertain-

25 ties.

Early versions of satellite SO$_2$ data products suffered from local biases caused by imperfect instrument calibration as well as from, for example, forward model simplifications (Fioletov et al., 2013; Yang et al., 2007). The PCA algorithm-based data set is practically free from such local biases. Nevertheless, we applied local bias correction to make possible estimation of point sources emissions in the areas with elevated background SO$_2$, such as north-eastern China or the eastern

30 U.S. For such areas, the average SO$_2$ VCD for the area located between 30 and 90 km upwind from the source for small sources (vs. 50 and 150 km for medium sources and 100 to 300 km for large sources) was used as the estimate of the bias and was subtracted from all data. Only days with wind speed greater than 4 km h$^{-1}$ were used for the bias calculation. The biases were estimated and removed for each year separately.



Several approaches were used to attribute an OMI $SO_2$ hotspot to a known source. The detected hotspots were compared to publicly available lists of known $SO_2$ emission sources. Web sites, such as http://globalenergyobservatory.org, http://www.industcards.com, and http://enipedia.tudelft.nl, were used to identify power plants and other industrial sources. Lists of smelters were available from http://mrdata.usgs.gov/copper and http://www.mining-atlas.com. In addition, the list of

Sulphuric acid-producing factories from http://www.sulphuric-acid.com was also used as such factories often utilize $SO_2$ produced by other industrial sources. It should be noted, however, that the above web resources might be outdated, incomplete, or contain incorrect coordinates. Google Earth imagery was used to verify the latter. Lastly, information on volcanic sources was obtained from Smithsonian Institution Global Volcanism Program (http://volcano.si.edu), whose source catalogue is incorporated into Google Earth.

Volcanic eruptions can eject large amounts of $SO_2$ into the upper troposphere and lower stratosphere where they can travel long distances (Ialongo et al., 2015; Karagulian et al., 2010; Spinei et al., 2010; Theys et al., 2015). The vertical column densities in volcanic plumes can be hundreds of DU (Carn et al., 2003; Krueger et al., 2008, 2000; Theys et al., 2013), whereas $SO_2$ values seen in the vicinity of many sources detectable by OMI are just a few tenths of a DU. If high volcanic values are not screened out, they would corrupt the anthropogenic emission estimates. To eliminate cases of

contamination by transient volcanic $SO_2$ plumes, any days when at least 1% of OMI measurements within 300 km of the pixel being analysed exceeded a cut-off limit were excluded from the analysis. We examined several cases in 2009 when emissions estimates were affected by $SO_2$ from the Sarychev (48.08°N, 153.21°E) volcanic eruption. Even with a 15 DU cut-off limit, the emissions for that year were still overestimated by about 60 kt $yr^{-1}$. Lowering the cut-off limit to 3 DU reduced that number by half, but such a limit may affect the estimates of actual emissions in the absence of volcanic

interference. Accordingly, the cut-off limit was set based on emission strength. It was set to 5 DU for sources that emit less than 100 kt $yr^{-1}$, to 10 DU for sources that emit between 100 kt $yr^{-1}$ and 1000 kt $yr^{-1}$, and to 15 DU for sources with emissions above 1000 kt $yr^{-1}$. For most anthropogenic emission sites, typically only one to two months for the entire record are affected. The same cut-off limits were applied to remove high $SO_2$ values from explosive volcanic eruptions. This may lead to underestimation of volcanic degassing by the applied method, but cases of such high $SO_2$ values are typically monitored

on a case-by-case basis (see OMI daily $SO_2$ maps for volcanic regions at http://so2.gfsc.nasa.gov). The volcanic $SO_2$ screening procedure can be improved in the future.

### 3.2. Uncertainty analysis

An error budget for the OMI-based emission estimates was constructed and the results are summarized in Table 1.

They are subject to uncertainties from three primary sources. The first source of error are the inputs used in the determination of the AMFs. Following (McLinden et al., 2014), surface reflectivity, surface pressure, ozone column, and cloud fraction and pressure com-bine for an uncertainty of 18%. The uncertainty from profile shape is more difficult to evaluate. Here AMFs were recalculated for different $SO_2$ profile assumptions including (a) exponentially-decreasing number





densities to the top of the PBL, and (b) fixed $SO_2$ layers of 1, 1.5, and 2 km. The standard deviation of these variations, 18%, was used to define this uncertainty. The AMF calculations assumed a Lambertian surface, and the uncertainty from this assumption was estimated to be 10%. The impact of aerosols was examined by including a layer between the surface and the top of the boundary layer, scaled to the aerosol optical depth from a 0.5° × 0.5° gridded climatology (Hsu et al.,

2012). The uncertainty from aerosol was estimated by adjusting the optical depth by ±0.25 about its assumed mean value (to a minimum of 0 and a maximum of 1) and recalculating AMFs and this results in a change of 10%.The overall AMF uncertainty was then found to be 28%.

The second source of error is related to the estimation of the total $SO_2$ mass as determined from a linear regression. This included the contribution from random errors of OMI measurements as well as variability of the emissions themselves,

which is particularly large for volcanic sources. The latter is often linked to the emission strength and can be expressed as a fraction of the estimated emission. For large sources, we estimated that the value of this parameter is about 5%. The noise in OMI data determines the sensitivity limit of the emissions estimation algorithm. By analysing small sources, we estimated that uncertainties in annual emission estimates are about 11 kt (1-σ) for 2005-2007 and about 16 kt for the following years as the row anomaly reduced the number of reliable OMI pixels. These values are lower in the tropics (6-8

15 kt) and higher at middle and high latitudes (~20 kt). The overall impact of this source of error is estimated to be 10 to 20 kt yr$^{-1}$ plus 5%. Due to its statistical nature, this source of error depends on the number of observations under low cloud amounts, which varies from site to site. Related to this are random errors in the ECMWF wind speed and direction, which were quantified by introducing random errors into real winds and determining how they impacted emissions (6%). Also, the error that results from a height offset was estimated by changing the height of the winds that were used by 500 m (20%).

The final source of uncertainty is from the fitting procedure. The use of prescribed values of σ and τ may not be optimal for a particular site. Their errors were discussed in section 3 and we can estimate the total uncertainty from the errors of the τ and σ values to be about 35%. All of these sources of uncertainty are summarized in Table 1. It should be noted that the third and largely the first sources of uncertainty are related to site-specific conditions and can be considered as systematic. They introduce a scaling factor in estimated emissions that affects absolute values, but not relative year-to-year

changes in emissions. Also, the choice of background and fitting regions also has a small impact on the emissions. Varying both of these by ±20 km led a 13% difference in estimated emissions.

## 4. NASA PCA vs. BIRA DOAS data sets

The retrieval algorithm itself could also be a source of uncertainty as the same spectral measurements could be processed in different ways and produce different $SO_2$ columns. As mentioned already, the previous NASA PBL algorithm

had random errors that were twice as high as the PCA algorithm as well as local biases and other artifacts. We compared the two state-of-the-art $SO_2$ data sets produced using the NASA PCA (Li et al., 2013) and BIRA DOAS (Theys et al., 2015) algorithms to evaluate possible uncertainties due to imperfections in the retrieval algorithms. To do this, we estimated




emissions for all catalogue sources using outputs from the two algorithms. The data were processed in exactly the same way for both datasets and the same data filtering was applied.

For NASA PCA data, the standard data product is based on the use of a constant AMF=0.36 to convert slant column density to VCD. The BIRA DOAS algorithm uses a different wavelength range and a constant AMF=0.42 corresponding to the same conditions (summertime eastern U.S.) as in the PCA data. Also, the two algorithms used $SO_2$ absorption spectra measured at different temperatures (283 K for PCA and 203 K for DOAS); the use of these different spec-tra creates a 19% difference in retrieved values. All of these differences were taken into account in the comparison.

We found that for most of the sites, the PCA and DOAS algorithms produced very similar results as illustrated by Figure 3a, where the 2005-2007 mean $SO_2$ distribution near the Bowen power plant (34.13°N, 84.92°W), USA, is shown. There are, however, some differences over regions of regionally elevated $SO_2$. As an extreme case, Figure 3b shows the 2005-2007 mean $SO_2$ distribution near Yangluo (30.69°N, 114.54°E), North China Plain, where the difference between mean PCA and DOAS values is about 0.5 DU. The difference appears as a large-scale bias and the bias correction procedure, described in section 3, removes it (Figure 3c). While the bias be-tween the PCA and DOAS data requires further investigation, it has practically no impact on the emission estimates. Figure 3d shows a scatter plot of emissions estimated from PCA and DOAS data for 2005-2007 for the roughly 500 sites analysed in this study. The correlation coefficient between the two data sets is 0.992. The slope of the regression line varies slightly from region to region, but remains within the 0.95 to 1.05 range, i.e., the emission estimates from the two algorithms agree to within 5%.

## 5. Source types

The anthropogenic emissions sources can be categorized in different ways by fuel type, by economic sector, region, or by their combinations. Our study focused primarily on single point sources, and the classification presented here is based on the four types of the largest point sources that can be monitored from space. These include fossil-fuel-burning power plants, e.g., near Johannesburg, South Africa, non-ferrous metal smelters such as the ones at Norilsk in northern Russia, and various oil and gas industry-related sources that can be seen, for example, in the Persian Gulf region, as illustrated by Figure 1. This classification is not always precise, as sources of different types could be collocated. Volcanic sources are also included in our classification, but are not the main focus of this study.

### 5.1 Coal- and Oil-fired Power Plants and Other Fuel-Combustion Sources

Coal-fired power plants and other coal-burning facilities are the most numerous type of $SO_2$ emission point sources seen by OMI. They are responsible for a majority of $SO_2$ emissions from China (Lu et al., 2011) and account for nearly all emission sources seen by OMI in the U.S., India, and Europe. $SO_2$ emission strength and detectability by satellite instruments depend on the sulphur content in the fuel and the extent to which sulphur in flue gas is captured by



desulphurization devices. For example, the $SO_2$ emission-factor ratio between power plants in southern and north-ern Greece is 25:1 (Kaldellis et al., 2004). While OMI clearly detected $SO_2$ emissions from the Megalopolis power plant (37.42°N, 22.11°E) in southern Greece in 2005-2007, $SO_2$ signals from the Aghios Dimitrios power plant (40.39°N, 21.92°E) and other power plants in northern Greece (Kardia, Ptolemadia, and Amyntaio, all located within 30 km of Aghios Dimitrios) were much weaker. The total capacity of the four power plants in northern Greece is 4000 MW vs. 850 MW for the Megalopolis power plant. However, OMI-based emission estimates for 2005-2007 for these sources are 76 kt $yr^{-1}$ and 384 kt $yr^{-1}$, respectively, for an OMI-estimated emission-factor ratio of about 24:1, i.e., similar to the value reported by (Kaldellis et al., 2004).

The installation of $SO_2$ scrubbers or a fuel switch to natural gas leads to a substantial reduction in $SO_2$ emissions that can be also confirmed by OMI. The steep decline in OMI mean $SO_2$ values in the vicinity of large U.S. coal-fired power plants over the period 2005 to 2010 was discussed previously (Fioletov et al., 2011). Other, similar examples are available for power plants in Spain and southern Greece, where high $SO_2$ values were seen in 2005-2007 but have declined since then. Changes in emission levels can be successfully traced from OMI-based emission estimates as illustrated in Figure 4, where time series of reported and OMI-estimated annual $SO_2$ emissions for the Megalopolis power plant, Greece, the Teruel-Andorra power plant (41.00°N, 0.38°W), Spain, and the Bowen power plant (34.13°N, 84.92°W), USA, are shown. The reported emission data were obtained from the European Pollutant Release and Transfer Register (http://prtr.ec.europa.eu), the Spanish Register of Emissions and Pollutant Sources (http://www.en.prtr-es.es), and the U.S. Environmental Protection Agency (http://www.epa.gov), respectively. All three sites had similar reported emissions (190-240 kt $yr^{-1}$) in the 2005-2007 period, but their reported emissions have declined to less than 50 kt $yr^{-1}$ after 2011. OMI-based emission estimates capture relative changes in the emissions well for these three sites. Estimated and reported emissions agree within 40 kt $yr^{-1}$ or ~20% for the Teruel-Andorra and Bowen power plants, but the OMI estimates for the Megalopolis power plant are more than 50% higher than the reported values. In OMI mean $SO_2$ maps (not shown), the Megalopolis signal is much stronger in the 2005-2007 period than the signals of the two other sources, and therefore it was expected that OMI-based estimates would produce substantially higher emission estimates for Megalopolis than for Teruel-Andorra and Bowen. Moreover, the Megalopolis $SO_2$ signal was clearly seen in OMI data in 2010, whereas according to the European inventory, it should be about 50 kt $yr^{-1}$, i.e., close to the OMI sensitivity limit. More research is required to determine the reason for this discrepancy, specifically whether the OMI $SO_2$ values over Megalopolis were too high (due, for example, to the use of an incorrect AMF value) or the reported emissions were somehow underestimated.

Combustion of fuel oil with high sulphur content can also produce strong $SO_2$ signal seen by OMI. As an example, Figure 5a shows the OMI $SO_2$ distribution near Havana, Cuba for the 2005-2007 period. In Cuba, fossil fuels supply nearly 92% of the total generated electricity and, for the most part, these are fuel oils with high (5%-7%) sulphur content (Turtós Carbonell et al., 2007). Three large oil-burning power plants are located near Havana. The Este de la Habana power plant (300 MW) is located in Santa Cruz to the east of Havana. The Maximo Gomez power plant (450 MW) is located in Mariel to the west of Havana. They emit about 76 and 98 kt $yr^{-1}$ (in 2003) of $SO_2$, respectively, as discussed by (Turtós Carbonell et





al., 2007). The distance between these first two plants is about 85 km. The third station, the Antonio Guiteras power plant (330 MW), is located 45 km to the east of Santa Cruz. As it uses the same type of domestic oil, it is expected that the $SO_2$ emission rate will be similar to that of the two other power plants, or close to 80 kt yr$^{-1}$ based on its power output.

We can use these three sources to illustrate how the algorithm described in section 3 estimates emissions for sources located in close proximity. The wind rotation procedure clearly demonstrates that upwind $SO_2$ values are lower than downwind values (Figure 5 b, c, and d). If there is a secondary source in the area at a distance R from the source, it manifests as a ring of elevated $SO_2$ values with radius R due to the wind rotation procedure. As the total mass is pre-served, the amplitude of the $SO_2$ signal would decline proportionally to 1/R. If the distance be-tween the two sources is small, they appear as one source, but if the distance is large, then 1/R is smaller and the second source becomes less visible and contributes less to the emission estimate. After the wind rotation is applied, the signal from Mariel looks weaker than from Santa Cruz, as the Este de la Habana and Antonio Guiteras power plants appear as a single source. Emission estimates (with a constant lifetime and spread) for 2005 produce a value of 83 kt yr$^{-1}$ for Mariel that is close to the reported value of 98 kt yr$^{-1}$ in 2003. If the fitting is done for source locations at Santa Cruz or Guiteras, however, the emission estimates for 2005 are 123 or 146 kt yr$^{-1}$, respectively, or close to the sum of the 2003 emissions from these power plants (~156 kt yr 1). This may suggest that for the OMI pixel size and the approach used in this study, sources located within about 50 km of one another will be interpreted as a single source with total emissions close to the sum of their emissions. However, pairs of sources can be distinguished as individual sources if the distance between them is greater than 80-100 km, although this limit would also depend on the emission strength and prevailing wind direction. To avoid double-counting emissions for regional averages, only two sites, Mariel (23.02°N, 82.75°W) and Guiteras (23.07°N, 81.54°W), are included in the catalogue. Similar choices have been made at other locations and are mentioned in the catalogue (see Supplement).

In the same vein, sources emitting less than 30 kt yr$^{-1}$ do not typically produce statistically significant signals in OMI data. If, however, there are several such sources in close proximity, their emissions can be seen by OMI. For example, the source labelled as Drax (53.74°N, 1.00°W), UK, is actually comprised of five coal-burning power plants and two oil refineries located within 50 km of Drax and emitting from 4 to 30 kt yr$^{-1}$ each. While the fitting procedure used here was optimized for single point sources, it still can produce reasonable estimates for the Drax source cluster: the 2005-2014 average estimated emissions are about 100 kt yr$^{-1}$ and the sum of reported emissions for those multiple sources for the same period is about 83-90 kt yr$^{-1}$ (depending on how missing data are treated). From our estimates, the uncertainties of annual emission estimates for Drax are, however, twice as large as for a single point source of the same strength.

Emissions from the iron and steel industry are also included in this category as the main source of $SO_2$ there is coal combustion. Examples of such sources in the catalogue include Baotou (40.66°N, 109.76°E), China, and Tata (22.79°N, 86.20°E), India, both of which are iron or steel factories where OMI data clearly show hotspots. In general, $SO_2$ hotspots are often located over industrial regions that include power plants and other sources and the attribution of a particular hotspot can be difficult. Most of the sources where the emission origin is not clear are included in this category.



## 5.2. Smelters

The smelting of sulphides of copper, nickel, zinc, and other base metal ores results in emissions of $SO_2$ that produce some of the largest point sources seen by OMI. When such ores are mined, they contain relatively small amounts of the desired metal, ranging from less than 1 percent for copper ore to up to 10 percent for lead and zinc ores. To increase the

metal content and to re-move other minerals, the ore is first grounded and concentrated. Concentrated copper ore typically contains 15% to 30% copper, 20% to 35% iron, 20% to 40% sulphur, and about 10%-15% of other minerals; lead concentrates contain 50% to 70% lead and 10% to 20% sulphur; zinc concentrates contain 60% zinc and 30% sulphur (United States General Accounting Office, 1986). Smelting the concentrated ore involves heating the concentrate to separate the desired metal from the sulphur and other materials. When heated, however, the sulphur in the concentrate oxidizes to

form sulphur dioxide.

$SO_2$ emissions from smelting depend on ore volume and sulphur content, and if $SO_2$ is not captured, emissions can be very substantial. For example, the Ilo smelter (17.50°S, 71.36°W), Peru, processes copper concentrate containing 33% sulphur from the Toquepala and Cuajone mines and produced 300 kt of copper per year (in 2001). About 30% of the $SO_2$ was converted into sulphuric acid, but 424 kt of $SO_2$ were still emitted (Boon et al., 2001). Using the previous version of the

OMI $SO_2$ data product, (Carn et al., 2007) estimated Ilo $SO_2$ emissions to be 300 kt yr$^{-1}$ by assuming a chemical lifetime of 1 day. Our new OMI-based estimates give larger values of about 1000 kt yr 1 in 2005-2006, but we assume a shorter 6-hour lifetime value. Regardless, the $SO_2$ signal from Ilo nearly disappears after 2007 as the smelter was modernized in February 2007 to satisfy new Peruvian environmental regulations.

The smelters in Norilsk (69.36°N, 88.13°E), Russia, combined, represent one of the largest, if not the largest,

anthropogenic $SO_2$ source that is clearly seen by satellites (Bauduin et al., 2014; Fioletov et al., 2013; Khokhar et al., 2008; Walter et al., 2012). The Norilsk annual copper and nickel production are about 350 kt yr$^{-1}$ and 130 kt yr$^{-1}$, respectively, with total $SO_2$ emissions of up to 1900 kt yr$^{-1}$ (http://www.nornik.ru). Independent estimates based on aircraft measurements in 2010 estimate its annual $SO_2$ emissions to be about 1000 kT yr$^{-1}$ (Walter et al., 2012). Our OMI-based estimates for Norilsk are between 1700 and 2300 kt yr$^{-1}$ with a 2005-2014 average of 2050 kt yr$^{-1}$.

Catalogue sites Chuquicamata (22.31°S, 68.89°W), and Caletones (34.11°S, 70.45°W) cor-respond to smelters in Chile that are among the world's largest, producing 500 and 400 kt of cop-per per year, respectively. However, they are located in the area where the South Atlantic Anoma-ly (SAA) significantly increases the noise in OMI retrieved data. Nonetheless, it is still possible to detect high $SO_2$ over these locations by averaging data over 5 to 10 years. Based on OMI estimates, emissions from Chuquicamata, and Caletones in 2005-2010 were 60 and 170 kt yr 1, respectively. These numbers

should be interpreted with great caution, though, since the uncertainties under the SAA are several times higher than outside the SAA. In recent years, emissions from Caletones have declined substantially, while no major change in emissions from Chuquicamata was seen.


As an illustration of OMI-based estimates of $SO_2$ emissions from smelters, in Figure 6 we have plotted time series of estimated annual emissions from four sources related to the smelting process. Highly elevated $SO_2$ signals over a copper smelter in Balkhash (46.83°N, 74.94°E), Kazakhstan, were seen not just by OMI, but also by other satellite instruments (Fioletov et al., 2013). The $SO_2$ signal from Balkhash was reduced substantially after 2008 when a sulphuric acid factory started to utilize emitted $SO_2$. For many years the Flin Flon copper and zinc smelter (54.77°N, 101.88°W) was one of the largest $SO_2$ emission sources in Canada, releasing about 200 kt of $SO_2$ per year. In 2010 operation of the smelter was stopped and no appreciable emissions are seen afterwards from that source.

We also included sources related to gold mining operations in the "smelter emissions" category. Figure 6 also shows a time series of annual $SO_2$ emissions from the Gidji gold roaster (catalogue site Gidji, 30.59°S, 121.46°E), Australia, which was designed for the roasting of refractory sulphide concentrate (Department of Environment and Conservation, 2006). Roasting the concentrate oxidizes the sulphide particles (pyrite) to iron oxide(s), making them porous so that the gold can be removed. Gidji is one of the largest $SO_2$ emission sources in Australia, with annual emissions of about 140 kt yr$^{-1}$. Total $SO_2$ emissions from the region around Gidji are even higher, about 200 kt yr$^{-1}$, due to two other large sources, the West Kalgoorlie nickel smelter and the Kanowna Belle Kalgoorlie gold mine, with emissions of about 30 kt yr$^{-1}$ each (Department of Environment and Conservation, 2006) that are located within 15 km. OMI-based estimates show a nearly constant level of annual emissions of about 160-180 kt yr$^{-1}$ in the 2005-2009 period, i.e., close to the total emissions from the three sources in the area.

The fourth source shown in Figure 6, Karabash (55.47N, 60.20E), is one of the oldest and largest copper smelters in Russia. It was closed in the early 1990s, but then re-opened in 1998. According to available information on $SO_2$ emissions (references in (Kalabin and Moiseenko, 2011)), emissions from Karabash in 2005 and 2006 were about 40 and 30 kt yr$^{-1}$, respectively. The OMI-estimated emissions for these two years were about 60 kt, i.e., higher by 20-30 kt, but within the uncertainty of the method. In the following years, the reported emissions declined further (Kalabin and Moiseenko, 2011) to just 5 kt yr$^{-1}$ in 2008. Instead, according to OMI, they in-creased to 300 kt yr 1 in 2014, making Karabash one of the largest anthropogenic $SO_2$ sources in the world. There could be some contribution from the nickel smelter in Ufaleynikel (56.05°N, 60.26°E), which is located just 60 km to the north, but estimated emissions from that source were lower in recent years than from Karabash. The reason for the discrepancy between reported and OMI-estimated emissions is thus not clear and should be investigated further.

Most of $SO_2$ sources related to smelting are associated with copper, nickel, and zinc smelting. However, there are several other types of $SO_2$ emissions from ore processing. OMI data show a clear $SO_2$ hotspot over Jamaica (listed in the catalogue as Manchester, 18.08°N, 77.48°W). It appears that the processing of high-sulphur bauxite for aluminium production is the main source of $SO_2$ air pollution in Jamaica. In 1994, it was responsible for 60% of about 100 kt yr$^{-1}$ emissions (http://www.nepa.gov.jm/regulations/RIAS-Final-Report-Technical-Support-Document-for-RIAS.pdf). The mean OMI-estimated emissions from Manchester for 2005-2007 were 112 kt yr$^{-1}$.





Iron refining activities are another source of $SO_2$. The Kostomuksha (64.65°N, 30.75°E), Russia iron mine and ore dressing mill is an example of such a source that is included in the catalogue. This site can also be used as an illustration of the sensitivity limits of our OMI-based estimates. The reported emissions are about 30-35 kt $yr^{-1}$ (Lehto et al., 2010; Potapova and Markkanen, 2003). The mean OMI-estimated emissions were 51 kt $yr^{-1}$ in 2005-2007 with a statistical uncertainty for the three-year average of about 15 kt $yr^{-1}$ (2-$\sigma$). The site is located at high latitude where observation conditions are difficult, and the OMI $SO_2$ emissions estimates are just above the limit of detectability. However, as there are no other sources in the vicinity, the origin of the emissions can easily be identified.

### 5.3. Oil and Gas industry

Oil refineries are another major source of $SO_2$ emissions. A variety of processes or operations in an oil refinery may produce $SO_2$ emissions, but three common refinery operations produce significant $SO_2$ emissions (Bingham et al., 1973). The first is catalyst regeneration. Catalysts used in catalytic crackers lose some of their activity after extended use and must either be regenerated or replaced. The regeneration process consists of oxidizing coke, which forms on the catalyst during cracking, to carbon monoxide. During regeneration, sulphur and sulphide deposits that also ac-cumulated on the catalyst are oxidized to $SO_2$. The second operation is hydrogen sulphide ($H_2S$) flaring. Many refinery processes produce off-gases that contain $H_2S$. All plants strip the $H_2S$ (usually in excess of 95 percent) from the off-gases before they are burned in process heaters and boilers. If the refinery does not convert the stripped $H_2S$ to sulphur, then the $H_2S$ stream is flared to the atmosphere and produces large amounts of $SO_2$. The third operation is fuel combustion. Much of the fuel required by refinery process heaters and boilers is produced by the refinery itself. Low-value distillate and residual oils with relatively high sulphur concentrations are often used for this purpose. While $SO_2$ can be removed for all three of these operations, the cost of the removal increases very rapidly as a function of the degree of emission reduction (Bingham et al., 1973). This is one reason why emission factors for $SO_2$ vary greatly from region to region. For example, the $SO_2$ emission factor for oil refineries in Iran was 119 times higher than in the U.K. (Karbassi et al., 2008).

As an example of an oil refinery-related source, Figure 7 (top) shows the $SO_2$ distribution in the vicinity of the Valero refinery (12.43°N, 69.90°W), Aruba in the Caribbean Sea. It is an isolated point source where persistent easterly trade winds form a clear pattern of the downwind $SO_2$ distribution. The Valero Aruba refinery processed lower-cost heavy sour crude oil (high sulphur content) and produced a high yield of finished distillate products with a total capacity of about 235,000 bpd. It was shut down temporarily in July 2009 due to poor economic conditions (Oil and Gas Journal, v. 107, issue 34, p. 10, 2009), reopened in late 2010, closed again in March 2012, and then converted to a products terminal (Oil and Gas Journal, v. 110, issue 9A, p. 13, 2012). Maps of the mean $SO_2$ distribution near Aruba for different periods (Figure 7 top) and an 11-year emission time series (Figure 7 bottom) show that OMI-estimated values track these changes in refinery activities well. Two more catalogue sources are also shown in Figure 7. The second source is the Paraguaná Refinery Complex (11.75°N, 70.20°W), Venezuela, one of the world's largest refinery complexes (940,000 bpd). The $SO_2$ signal



from the third source, Isla refinery (12.13°N, 68.93°W), Curacao (320,000 bpd), is likely responsible for the small $SO_2$ hotspot to the east of Aruba. Note that the Valero Aruba refinery capacity was the smallest of these three sources whereas the emissions were the largest, suggesting a role for the fuel type (as well as emission controls).

The number of oil and gas industry-related $SO_2$ emission sources is particularly large in the Middle East. Oil refineries and power plants are often collocated in this region as in Isfahan (32.79°N, 51.51°E), Iran (370,000 bpd capacity) and Rabigh (22.67°N, 39.03°E) (400,000 bpd) and Jeddah (21.44°N, 39.18°E) (100,000 bpd), Saudi Arabia. Such collocation makes the attribution of source type in the absence of additional information very problematic. Many hotspots in the Middle East, however, are not associated with large individual facilities but are collocated with oil fields as shown in Figure 8 (sources Dehloran, Ahvaz, and Feridoon). Flaring in these oil fields is the likely source of these $SO_2$ emissions. The $SO_2$ is emitted as a result of oxidation of $H_2S$ from flaring of $H_2S$-rich off-gas (Abdul-Wahab et al., 2012). For example, $SO_2$ emissions from flaring of sour gas (rich in $H_2S$ and mercaptans) from Kuwait alone are up to 100 kt $yr^{-1}$ (AL-Hamad and Khan, 2008). Emissions depend on the composition of the flared gas and could be very different from one oil field to another. Information about $SO_2$ emissions from flaring is very sparse, however, and such sources are often not included in major emission inventories.

Natural gas refining is a process of removal of contaminants, including sulphur compounds, before distributing it to consumers. The source in the upper-right corner in Figure 8 is the Khangiran gas refinery (36.47°N, 60.85°E), Iran, where strong $SO_2$ emissions are related to the gas refining process. The Shahid Hashemi-Nejad (Khangiran) refinery is one of the most important gas re-fineries in Iran and processes natural gas supplied by the Mozdouran gas fields. The Khangiran gas refinery normally burns off 25,000 m3 $h^{-1}$ gas in flare stacks. Although some sulphur is captured by sulphur recovery units, there is still a sizable fraction of $H_2S$ in flare gas (Zadakbar et al., 2008). It is expected, however, that there will be a decline in $SO_2$ emissions from this facility as "in March 2013, the first phase of the project to cut gas burning in flares at Khangiran refinery got underway" (http://theiranproject.com/blog/2013/06/27/khangiran-refinery-produces-50-mcm-of-natural-gas-per-day). Note that information on Khangiran $SO_2$ emissions is not included in any of the major emission inventories.

**5.4. Volcanoes**

OMI data are widely used to monitor volcanic $SO_2$ emissions from both eruptions and de-gassing of individual volcanoes (Bluth and Carn, 2008; Campion et al., 2012; Carn et al., 2004, 2008, 2013, 2016; Krotkov et al., 1997; McCormick et al., 2012, 2014). Satellite monitoring of $SO_2$ emissions from volcanoes, however, has numerous issues such as limited instrument sensitivity to volcanic plumes at low altitudes and interference from volcanic ash (McCormick et al., 2013), although the latter is less significant for the volcanic degassing emissions that are the focus of this work. Albedo effects from snow-covered volcanic cones and uncertainty of the height of the volcanic plume can also contribute to emission uncertainties. Furthermore, the present NASA PCA $SO_2$ data product is optimized for boundary-layer $SO_2$ vertical



distributions, which is not always suitable for volcanic degassing sources. It is thus important to remember that for this study we have corrected PCA data using altitude-dependent AMFs as described in section 2.

As an illustration of OMI-based estimates of $SO_2$ emissions from volcanoes, Figure 9 shows $SO_2$ emissions from four volcanoes in Japan. They are probably the most monitored volcanoes in the world (Mori et al., 2013), with information

on their activity and $SO_2$ emissions regularly published by the Japan Meteorological Agency (JMA, http://www.data.jma.go.jp/svd/vois/data/tokyo/STOCK/souran_eng/souran.htm). There is a very good qualitative agreement between the JMA $SO_2$ emission measurements and our OMI-based estimates: periods of low and high $SO_2$ emissions were captured by OMI very well and they clearly show similar long-term tendencies in volcanic $SO_2$ fluxes. Quantitatively, seasonal mean emission estimates from OMI can differ from JMA estimates by 50%, but the days sampled by the two

methods could be very different as satellite information is not available on cloudy days.

Improving satellite retrieval and data analysis algorithms also allows remote monitoring of emissions from volcanoes that were not detectable in the past. For example, (McCormick et al., 2013) mentioned that emissions from Stromboli (38.79°N, 15.21°E), Italy, were not detected in the previous OMI data set due to low $SO_2$ fluxes and their proximity to much stronger $SO_2$ emissions from Mount Etna. However, when the new PCA version is used and the new

emissions estimation algorithm is applied, the Stromboli signal is clearly detectable and this source is included in the catalogue. The OMI-estimated 2005-2014 mean annual $SO_2$ emission for Stromboli is about 60 kt yr 1, which is not too different from reported emissions of about 200 t day$^{-1}$ or 70 kt yr$^{-1}$ (Burton et al., 2009). (McCormick et al., 2013) also discussed volcanic emissions from Mount Etna, Italy and Popocatépetl, Mexico, quoting $SO_2$ emission levels from 600 to 1300 kt yr$^{-1}$ and from 900 to 1900 kt yr$^{-1}$, respectively. Our OMI-based estimated annual mean emissions for Mount Etna

range from 530 to 1200 kt yr$^{-1}$, i.e., similar to the values provided by (McCormick et al., 2013). Our OMI-based estimated $SO_2$ emissions for Popocatépetl, on the other hand, range from 300 to 1200 kt yr$^{-1}$, i.e., lower than the values from (McCormick et al., 2013) but in general agreement with an estimate of 2.45±1.39 kt day$^{-1}$ or 900±500 kt yr$^{-1}$ by (Grutter et al., 2008).

For many remote volcanoes, satellite-based estimation is the only feasible source of emissions information. For

example, the catalogue includes the first $SO_2$ emission estimates for Michael (57.80°S, 26.49°W) and Montagu (58.42°S, 26.33°W) volcanoes, South Sandwich Islands (UK), and several volcanoes in the Aleutian Islands (Alaska, USA), which are known to be active (Gassó, 2008; Patrick and Smellie, 2013) but for which no information is available in major emission databases.

Detailed comparison of OMI-estimated emissions with the available information about volcanic $SO_2$ fluxes is

beyond the scope of this paper. Rather, the main goal here is to introduce the catalogue and to provide a first version of estimated emissions for these important natural sources. It is expected that more accurate OMI-based volcanic emissions estimates will be available when the improved PCA volcanic $SO_2$ data products are developed with assumed $SO_2$ vertical profiles more suitable for volcanic sources.



## 6. The catalogue

A total of 491 continuously emitting point sources releasing from about 30 kt to more than 4000 kt of $SO_2$ per year have been identified using OMI measurements and have been grouped by country and by source type as follows: power plants (297); smelters (53); sources related to the oil and gas industry (65); and volcanoes (76 sources) (see Figure 10 for
their locations). The catalogue file is an MS Excel file that contains the site coordinates, source type, country, source name, and other information and is available as a Supplement to this study. Note that sites in the catalogue are labelled by simple names to make it easy to search the catalogue and to display them in Google Earth applications. Where possible, we used the actual facility or volcano name; other-wise, the sites were labelled by the name of the nearest town. In cases of multiple sources, we tried to assign the site coordinates to the largest source. Some additional information such as the location of
nearby secondary sources is provided in the "Comment" column.

In addition to the site location, country, source name, and source type, the Supplement file also contains estimates of annual emissions and their uncertainties for the 2005-2014 period. As an illustration, Figure 11 shows mean annual emissions for two multi-year periods, 2005-2007 and 2012-2014. The largest sources are volcanoes, although the Norilsk smelter and a cluster of power plants in South Africa are not far behind. Relative changes between the two periods are
shown in the bottom panel of Figure 11. Blue dots indicating a decline in emissions are numerous in the U.S., Europe, and China, and to a large extent they reflect recent installation of scrubbers on power plants or fuel switching (e.g., Fioletov et al., 2011; Klimont et al., 2013; Krotkov et al., 2015). Conversely, an increase of emissions over the same period as represented by red dots can be seen over India, Mexico, Venezuela, and Iran.

It should be mentioned that the attribution of the sources was done based on our best knowledge and may not
always be correct. As already mentioned in section 5, in some cases, there are several individual sources in close proximity and it is difficult to estimate contribution of each of them. For others, no definitive information was found on the source origin. While the emission estimation algorithm was developed for point sources, it works reasonably well when there are two or even more sources in close proximity (20-30 km) but with no other sources nearby. There are, however, some regions of China where sources are dense enough that it becomes difficult to apply the algorithm. In these instances, we
simply identified hotspots and included them in the catalogue to have a reasonable representation of the total emissions for such regions. These hotspots are labelled as "Area" sources in the catalogue (e.g., Liaoning, Wuan). This treatment can be improved in the future when more detailed information about the sources and the emissions from them become available. Such a database for China is under development (Liu et al., 2015).

## 7. Comparison with Emission Inventories

Emission estimates from OMI for individual sources can be further grouped by source type to study the contribution of different source types to total $SO_2$ emissions. Some of the results of this section have been presented in our previous study (McLinden et al., 2016), and here we provide additional information as well as a sensitivity to AMF study. Figure 12 shows



time series of total annual $SO_2$ emissions for the four primary source types: power plants; smelters; oil and gas industry sources; and volcanoes. As mentioned in section 5.1, installation of flue-gas scrubbers has substantially reduced emissions from many U.S., European, and Chinese coal-fired power plants, resulting in an overall decline in total emissions from that type of source. Total emissions from the world's largest metal smelting-related sources have also declined substantially

during the period of OMI operation as some of them have ceased operation temporarily or permanently (e.g., Ilo, Peru; Flin Flon, Canada), while others have installed scrubbers (e.g., La Oroya, Peru) or started to collect $SO_2$ for sulphuric acid production (e.g., Balkhash, Kazakhstan). In contrast, there were no significant changes in total emissions from oil and gas industry-related sources.

Correct assessment of total volcanic $SO_2$ emissions depends on the AMF value that is used. Estimated total

volcanic emissions are almost 40% higher for a constant AMF=0.36 than for an altitude-dependent AMF (Figure 12a and b) since many volcanoes have heights above 1000 m. Therefore, current PCA data products should be used with caution for volcanic sources. On the other hand, the inclusion of other factors such as albedo and the mean PBL height in the AMF calculations has little effect on the total volcanic emissions (Figure 12c). Note that the differences resulting from the three different ways to calculate AMF are much smaller, within about 10%, for anthropogenic sources.

Based on the estimates presented here, the total $SO_2$ emissions from all volcanic sources included in the catalogue accounted for about 25% of all OMI-based emissions in 2005 (Figure 12c). That fraction increased to 32% in 2014 due to a decline in emissions from power plants and smelters. Note, however, that numerous small sources (with annual emissions under ~30 kt) are not detected by OMI and therefore are not included in the total estimates. As a result, the total anthropogenic emissions are underestimated by OMI, but this should also be true for volcanic sources and hence may not

affect the ratio of the volcanic to total $SO_2$ emissions. However, the proportion of volcanic $SO_2$ emissions relative to the total will show a significant regional variation due to the geographic distribution of volcanic and anthropogenic sources.

Emissions from individual sources in the catalogue can also be aggregated into national or regional totals and then compared with the available "bottom-up" emissions inventories. This approach is different from the one used by (Krotkov et al., 2015, this issue), where regional averages were calculated first and then their temporal changes were studied. Figure 13

shows the temporal evolution of total OMI-based $SO_2$ emissions over the 2005-2014 period for 8 countries/regions, where emissions were summed over individual sources in each region after calculations using three different AMFs. Comparison of Figures 13 a-c demonstrate the impact of the AMF values on the resulting absolute emission levels. Note that the consideration of altitude has a noticeable impact on the estimates for South Africa as power plants there are located at 1500 m ASL. Accounting for albedo reduces emissions estimates for the Middle East by about 20%, as many sources there are

located in sand-covered areas with high albedos. On the other hand, accounting for albedo has the opposite effect on total emission estimates for Russia, with an almost 40% increase in emission estimates (compared to AMF=0.36) for Norilsk, the largest $SO_2$ source in Russia, which accounts for almost half of the total OMI-based emissions from that country (note that measurements with high albedo caused by snow are excluded from the analysis as discussed in section 2). For AMF=0.36, 2014 emissions from the Middle East were estimated to be the second-highest in the world after China, followed by India



and Russia (Figure 13a) with Russian emissions being nearly 50% lower than those from the Middle East. If site-specific AMF values are used, though, then SO$_2$ emissions from the Middle East and Russia become comparable and emissions from the India are just slightly lower.

According to Figure 13c, the most dramatic decline, about 70-80%, can be seen for U.S. sources. This is in line with estimates from bottom-up U.S. emission inventories (http://www.epa.gov/ttnchie1/trends/) and is largely the result of a combination of the installation of flue-gas scrubbers at some U.S. power plants, the closure of some older coal-fired power plants, and conversions of some power plants from coal to natural gas. A decline by 40-50% can be seen for the sum of all European sources. These estimates are also similar to these from OMI gridded data (Krotkov et al., 2015). By contrast a roughly 80% increase in emissions can be seen over India, although for some regions the increase is as large as 200% (Krotkov et al., 2015). The Middle East is the region with the largest SO$_2$ emissions after China, and these emissions are nearly constant. The estimated total annual SO$_2$ emissions for the Middle East are about 6 Mt, with Iranian sources contributing about half of the total. Mexico, Russia, and South Africa are among the largest SO$_2$-emitting countries and they show no obvious trends.

In addition, the recent global SO$_2$ bottom-up emissions inventory constructed by (Klimont et al., 2013) was compared with the OMI-based regional estimates developed in this study. (Klimont et al., 2013) is an extension to 2011 of previously published global SO$_2$ inventories (Smith et al., 2004, 2011), and it has group/regional annual emission tables in its supplement that are convenient for a comparison with OMI-based estimates. Average total SO$_2$ emissions for 2005-2011 by region estimated from OMI data and from this emission inventory (Klimont et al., 2013) and their ratios are presented in Table 2. As OMI does not detect small sources, OMI-based emission estimates should be lower than the actual emissions. The ratios of about 0.4-0.5 seen for the U.S. and Europe are therefore expected. For most of the 14 regions, the ratios are within the 0.5±0.15 range, meaning that the spread of the ratios is ±30%, i.e., even better than could be expected from Table 1. There are some exceptions, but they are likely related to the emission inventories rather than errors in the OMI-based emissions estimates. For example, very large values of the ratio for Mexico are probably caused by unreported emissions.

The 2005-2011 temporal evolution of these ratios (i.e., OMI-based emissions to bottom-up emissions) is shown in Figure 13 d-f. The ratios for the U.S., Europe, and India are nearly constant, although the actual emissions have changed very substantially. This suggests that the de-scribed OMI-based estimates can successfully capture at least relative changes in emissions. The ratios are also nearly constant for Middle East and South Africa. The ratios for Russia and Turkey (not shown) suggest some increase because their reported emissions have a negative trend, whereas OMI-based estimates are either constant (Russia) or increasing (Turkey). The largest increase in the ratios can be seen for Mexico. According to (Klimont et al., 2013), Mexican anthropogenic SO$_2$ emissions declined by 60% from 1.7 Mt yr$^{-1}$ to 0.7 Mt yr$^{-1}$ between 2005 and 2011. This is clearly not confirmed by OMI. Moreover, according to OMI, SO$_2$ emissions from just one source, oil fields in the Gulf of Mexico (19.40°N, 92.24°W) that are detected by several satellite instruments (Fioletov et al., 2013), are comparable in magnitude to the total reported Mexican emissions in 2011, but emissions from that off-shore source are not included in available emission inventories.



Figure 13a-c also shows that accounting for various factors in AMF calculations reduces the spread in the OMI-estimated to inventory-reported emission ratios. This may indicate that the adjustment we applied to the standard PCA data products corrects the data in the right direction and leads to the better agreement between estimated and reported emissions.

## 8. Summary

This study introduces the first space-based catalogue of $SO_2$ emission sources seen by OMI. A total of 491 point sources with annual $SO_2$ emissions ranging from about 30 kt yr$^{-1}$ to more than 4000 kt yr$^{-1}$ are included in the catalogue. Annual emission estimates and their uncertainties derived from OMI data are also provided for the period 2005-2014. Source types have been identified using available databases of anthropogenic and natural $SO_2$ sources. A total of 297 power plants, 53 smelters, 65 sources related to the oil and gas industry, and 76 volcanoes are included in this first version of the catalogue. It should be mentioned that simple attribution is not always possible because at some sites multiple different industrial sources are clustered in close proximity. Source identification from OMI data is particularly difficult in China, where point sources are numerous and are often located in clusters.

Two different versions of the OMI $SO_2$ data product, the NASA PCA algorithm-based version and the BIRA DOAS algorithm-based version, were tested. While large-scale biases are somewhat different, particularly over areas of elevated $SO_2$ levels, the emissions for point sources estimated from the two data sets are very similar, with a correlation coefficient above 0.99 and systematic differences within ±5%.

Statistical uncertainties (1-σ) of the emission estimates are approximately 10 to 20 kt yr$^{-1}$ plus 5%. The uncertainties caused by the retrieval algorithms including AMF values are estimated at 50-60%, but comparisons with reliable bottom-up inventories typically indicate agreement to better than 30% (based on the spread of the OMI estimated to reported ratios). For a number of sites that we have examined in this study, the OMI-based estimates of annual emissions show very good qualitative agreement, capturing changes in emission rates caused by scrubber installations and interruptions in facility operation as well as major changes in volcanic activity.

The emission estimation algorithm has been developed for point sources. If more than one source are located in close proximity, the emission estimation algorithm may not be able to distinguish between them. For sources with annual emissions of about 100 kt, other sources located within about 50 km are seen as a single source, while emissions for each source can be estimated separately if the separation distance is greater than 100 km.

The standard NASA PCA data product based on the summertime eastern U.S. conditions with AMF of 0.36 should be used with caution when absolute emissions for other regions are calculated. For example, accounting for elevation in AMF calculations reduces the total volcanic emissions seen by OMI by about 40%. Accounting for albedo variations changes emissions esti-mates for the Middle East and Russia: the emissions from the Middle East are almost twice as high as those from Russia for AMF=0.36, but they become comparable if albedo differences are accounted for. These dependencies demonstrate the need for a better estimation of AMFs for different regions.



Ratios between OMI-estimated and bottom-up reported annual emissions for most of the large countries and regions are within 0.5±0.15 limits. This was expected because OMI cannot estimate emissions from numerous small (<30 kt yr$^{-1}$) sources. These ratios for the U.S., Europe, India, Middle East, and South Africa are also fairly constant over time, suggesting that OMI can be used to trace regional emission trends. The ratio for Mexico is increasing, most likely due to in-complete
reporting of facility emissions, especially from off-shore oil and gas production.

The catalogue presented herein can be used for verification of $SO_2$ emission inventories and identification of missing sources. It can be also used to fill gaps in available inventories, particularly if there are no other sources of information, e.g., for remote volcanoes. Conversely, those sites for which reliable $SO_2$ emissions data are available can be used for OMI $SO_2$ data product validation. The catalogue could also be used for cross-validation of different satellite data
sources, similar to the comparison done for OMI, GOME-2, and SCIAMACHY (Fioletov et al., 2013). This could be particularly useful for cross-validation of new polar-orbiting satellite instruments such as TROPOspheric Monitoring Instrument (TROPOMI) (Veefkind et al., 2012), which is planned for launch on ESA's Sentinel 5 Precursor satellite in 2016, and the data from three new-generation geostationary satellites scheduled to be put into orbits over North America (TEMPO, http://tempo.si.edu), Europe (Sentinel 4), and Asia (Geostationary Environment Monitoring Spectrometer).
Lastly, it is expected that this catalogue will only be a first version, and it will be further updated, enhanced, and improved under NASA's "Making Earth System Data Records for Use in Research Environments" (MEaSUREs) $SO_2$ project (http://so2.gsfc.nasa.gov/measures.html).

**Acknowledgments.** We acknowledge the NASA Earth Science Division, specifically the Aura science team program, for
funding OMI $SO_2$ product development and analysis. The Dutch-Finnish-built OMI instrument is part of the NASA EOS Aura satellite payload. The OMI project is managed by KNMI and the Netherlands Space Agency (NSO). OMI PCA $SO_2$ data used in this study have been publicly released as part of the Aura OMI Sulphur Dioxide Data Product - OMSO2 and can be obtained free of charge from the Goddard Earth Sciences (GES) Data and Information Services Center (DISC, http://daac.gsfc.nasa.gov/).

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





**Table 1.** Uncertainty budget for annual emissions estimate

| Error category | Source | Magnitude | Note |
|---|---|---|---|
| Air mass factor | Variability | 18% | Considers random errors in cloud fraction, cloud pressure, surface albedo, surface pressure, and column ozone |
| | Variability | 18% | Profile shape |
| | Uncertainty | 14% | BRDF (10%) and aerosol (10%) |
| Mass | Linear fit | 5% | Statistical errors from the regression model |
| | Uncertainties in OMI retrieved values | 10-20 kt | |
| Lifetime and Width | Uncertainty | 35% | Used prescribed values may be different from actual ones. From sensitivity study (Section 3) |
| Local bias estimates | Uncertainty | 13% | Fitting limits |
| Wind-speed and direction | Variability | 6% | From random errors in wind speed (2 m/s) and direction (15°) |
| | Uncertainty | 20% | Systematic effect from taking the winds at the wrong height |
| Total | | 55% | For sources above 100 kt |
| | | >67% | For sources under 50 kt |



**Table 2.** Average total $SO_2$ emissions for 2005-2011 by region (kt yr$^{-1}$) estimated from OMI data and from emission inventories (Klimont et al., 2013), and the ratio of the OMI-based estimates to the inventory values.

| Country/Area | Number of OMI-identified sites | OMI-based estimates | Emission inventories | Ratio |
|---|---|---|---|---|
| Canada | 7 | 631 | 2071 | 0.30 |
| India | 46 | 3160 | 8410 | 0.38 |
| USA | 56 | 4288 | 10010 | 0.41 |
| Turkey | 7 | 783 | 1515 | 0.52 |
| China | 82 | 17060 | 31352 | 0.54 |
| Europe[*] | 35 | 3795 | 6911 | 0.55 |
| South Africa | 5 | 1550 | 2689 | 0.58 |
| Central Asia | 14 | 1292 | 2095 | 0.62 |
| Australia | 9 | 907 | 1391 | 0.65 |
| Russia | 29 | 4312 | 5344 | 0.82 |
| Central America | 16 | 1941 | 2298 | 0.87 |
| Ukraine[**] | 11 | 1254 | 1323 | 0.96 |
| Middle East | 42 | 5634 | 5509 | 1.02 |
| Mexico | 19 | 2348 | 1158 | 2.24 |

[*]Western and Central Europe

5    [**]Also includes Belarus





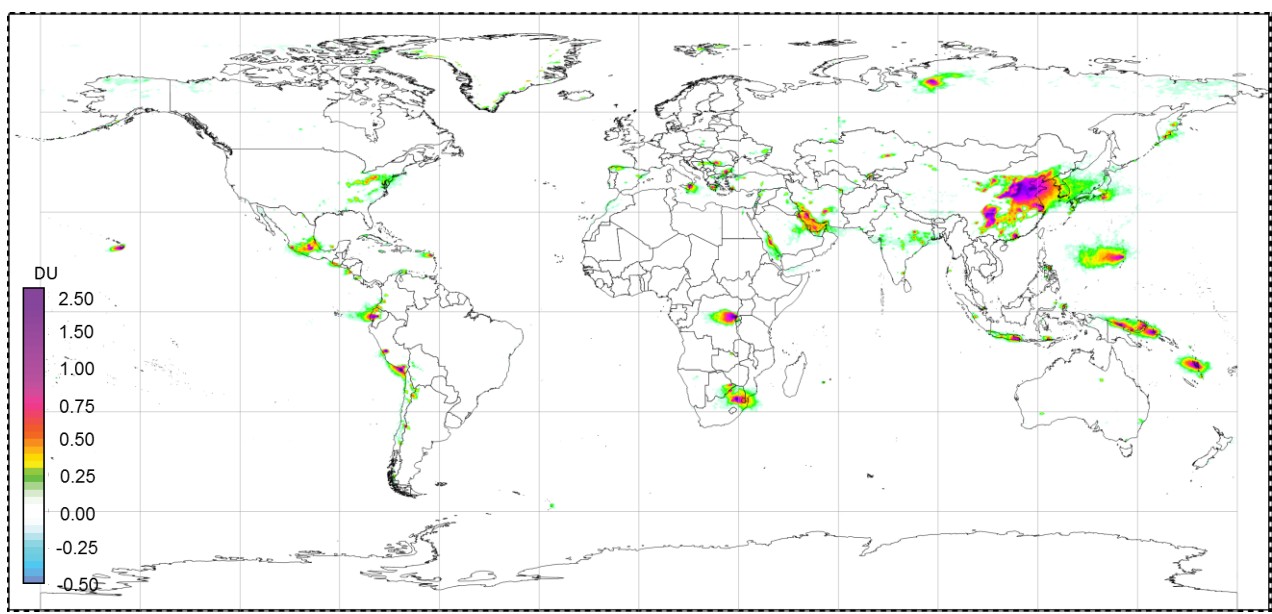

**Figure 1**. The global mean SO$_2$ distribution (in DU) map for 2005-2007. The area affected by the South Atlantic Anomaly is hidden.




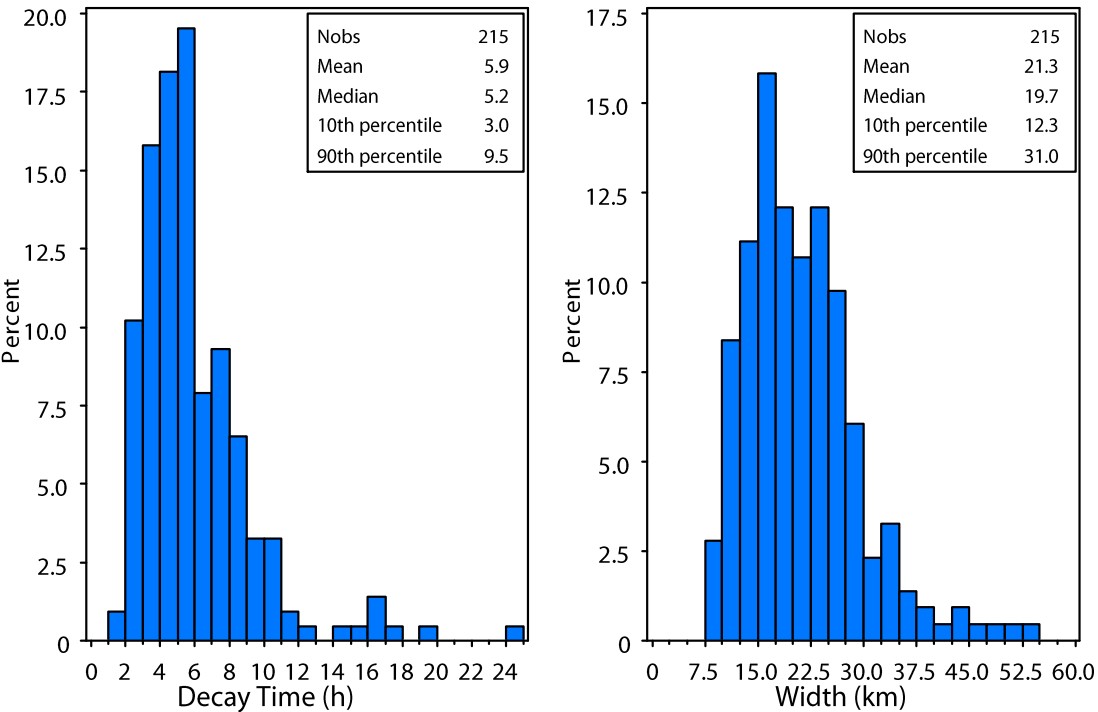

**Figure 2** The distribution of the estimated decay time ($\tau$) and plume width ($\sigma$) obtained from the fit of OMI data for the 2005-2007 period. Data from 215 catalogue sites that produced estimates of $\sigma$ and $\tau$ with small uncertainties were used for the plot. The main statistical characteristics of the distribution are also shown.

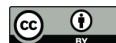



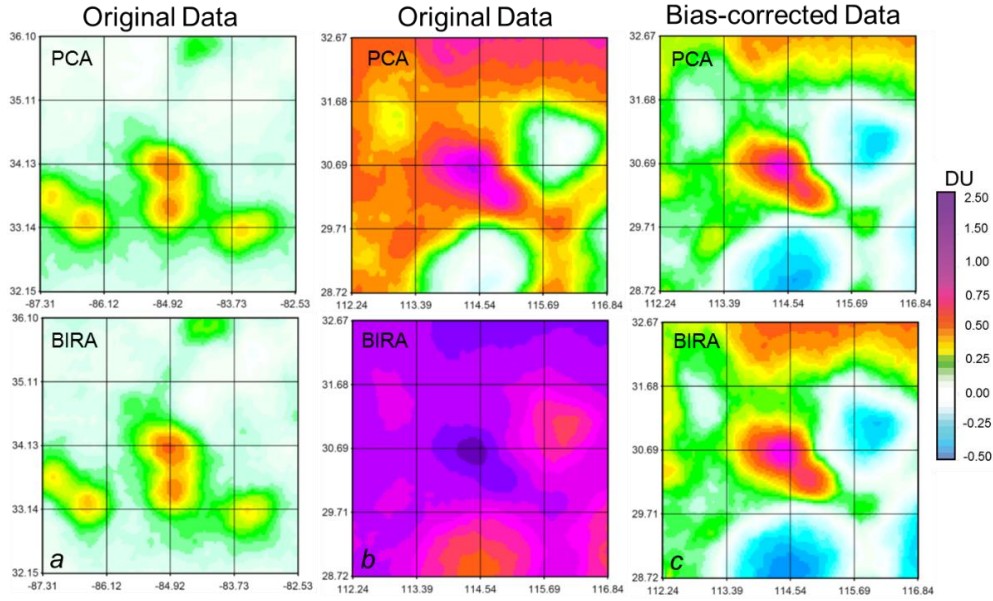

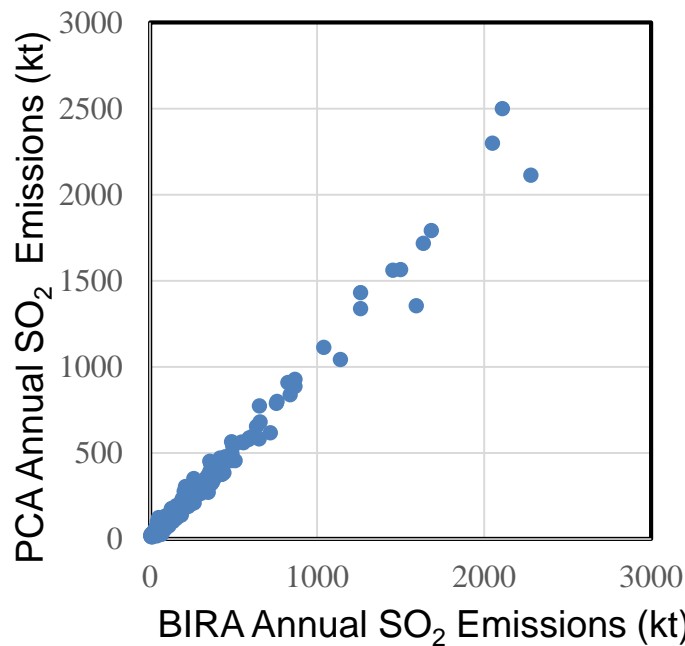

**Figure 3.** The 2005-2007 mean $SO_2$ VCD distribution (a) near Bowen power plant (34.13°N, 84.92°W), USA, and (b) near Yangluo (30.69°N, 114.54°E), China, from NASA PCA and BIRA DOAS data. (c) The mean $SO_2$ distribution near Yangluo with local bias removed as described in section 3. (d) The scatter plot of emissions estimated from PCA and BIRA DOAS data for 2005-2007 for about 500 sites analyzed in this study. The correlation coefficient between the two data sets is 0.992.



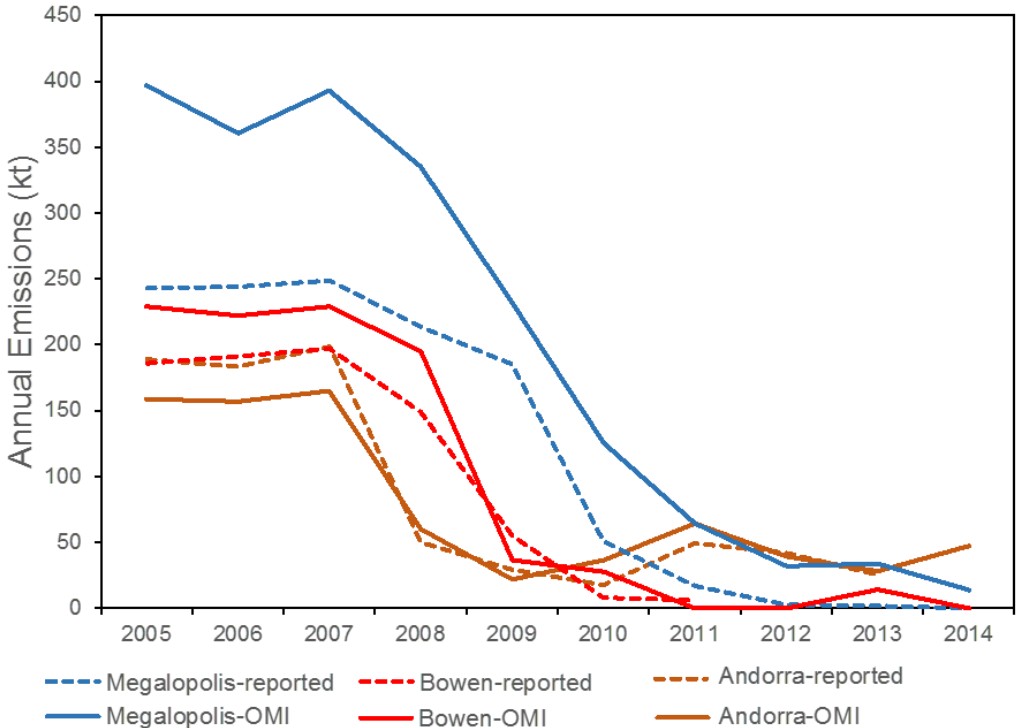

**Figure 4.** Time series of annual SO$_2$ emissions for Bowen, USA (red); Teruel-Andorra, Spain (brown); and Megalopolis, Greece (blue); power plants. Estimated from OMI and reported annual emissions for 2005-2014 are shown by the solid and dotted lines respectively.



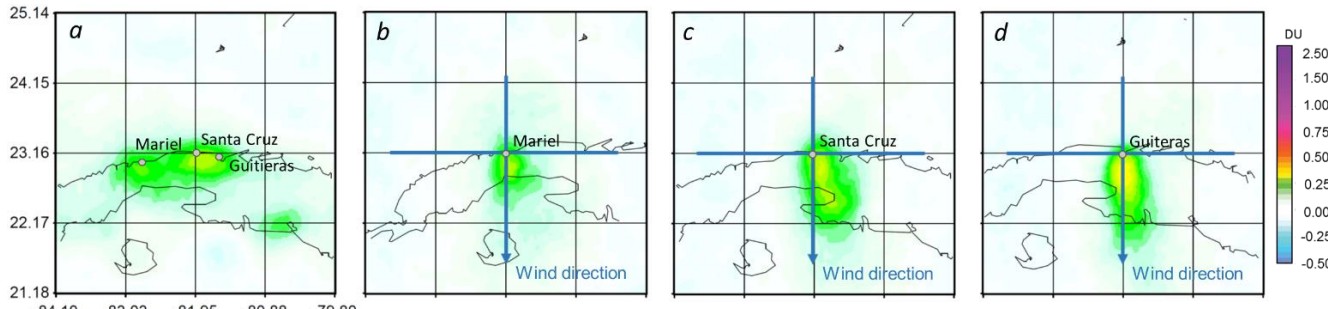

**Figure 5.** (a) Mean SO$_2$ distribution near Santa Cruz (Este de la Habana power plant), Mariel (Maximo Gomez power plant)
and the Antonio Guiteras power plant, Cuba, for 2005-2007. These power plants use Cuban domestic sulfur-heavy oil. Mean
SO$_2$ distributions near (b) Mariel, (c) Santa Cruz, and (d) Guiteras after the wind rotation procedure was applied to each site.




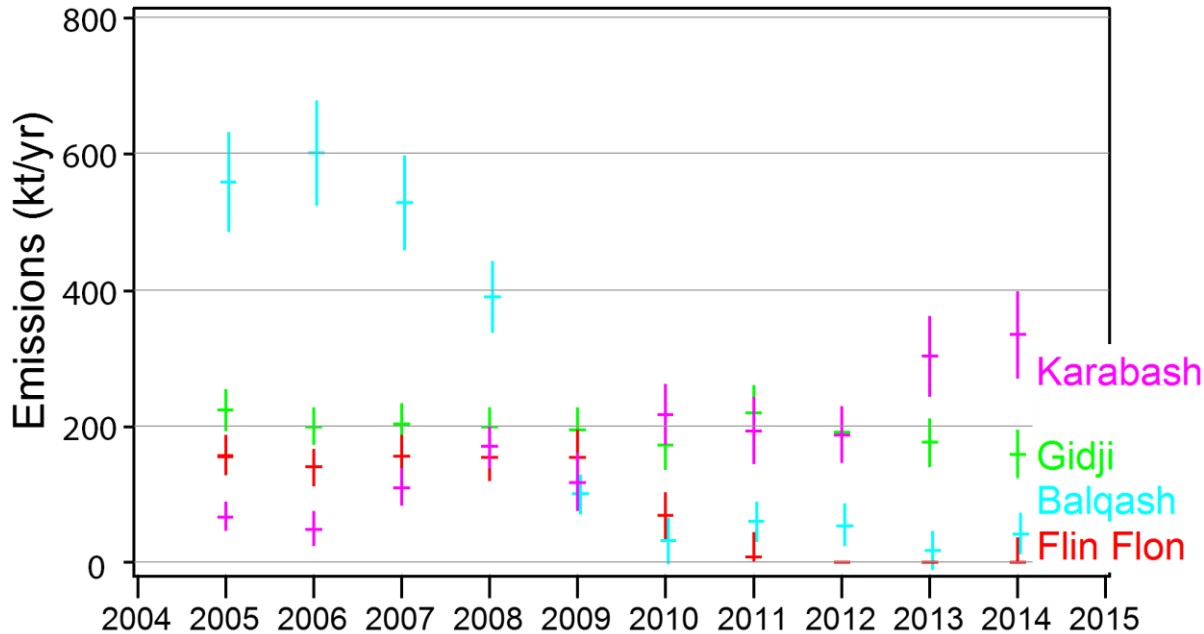

**Figure 6.** Estimated annual emissions from four smelters: Karabash, Russia; Gidji, Australia; Balqash, Kazakhstan; and Flin Flon, Canada, with 2-σ error bars. Note that emissions from Balkhash smelter have been reduced substantially after 2008 when a sulfuric acid factory started to utilize emitted $SO_2$. The operation of the Flin Flon copper and zinc smelter was stopped in 2010.



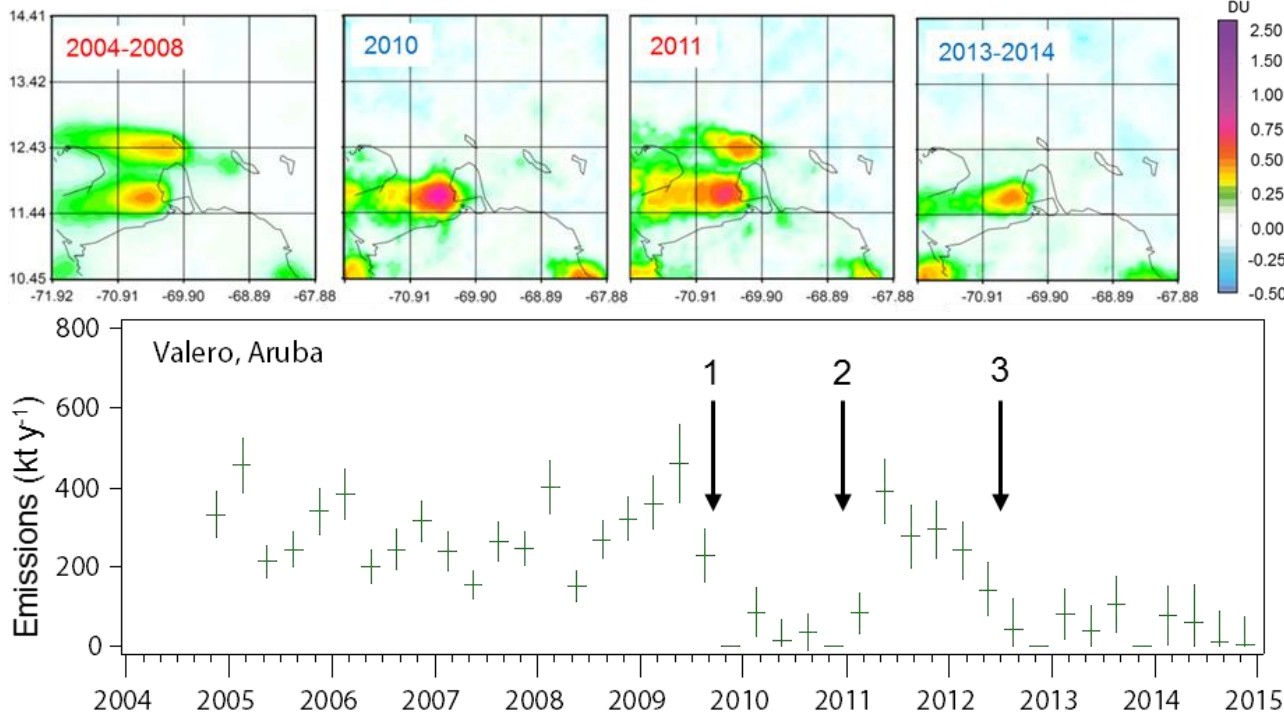

**Figure 7.** (top) Mean $SO_2$ maps near the Valero refinery, Aruba, located at the center, for different time intervals. Years when the refinery was operational are labeled in red, and years when it was idle are labeled in blue. The Paraguaná refinery complex, Venezuela, and Isla refinery, Curacao, are responsible for the $SO_2$ hotspots located south and east of Aruba respectively. (bottom) Time series of Valero $SO_2$ emission rates estimated from OMI. Each symbol represents an estimate for a 3-month period with 2-$\sigma$ error bars. The refinery was shut down temporarily in July 2009 (marked by arrow 1), reopened in late 2010 (arrow 2), and was then converted to a product terminal starting in March 2012 (arrow 3).

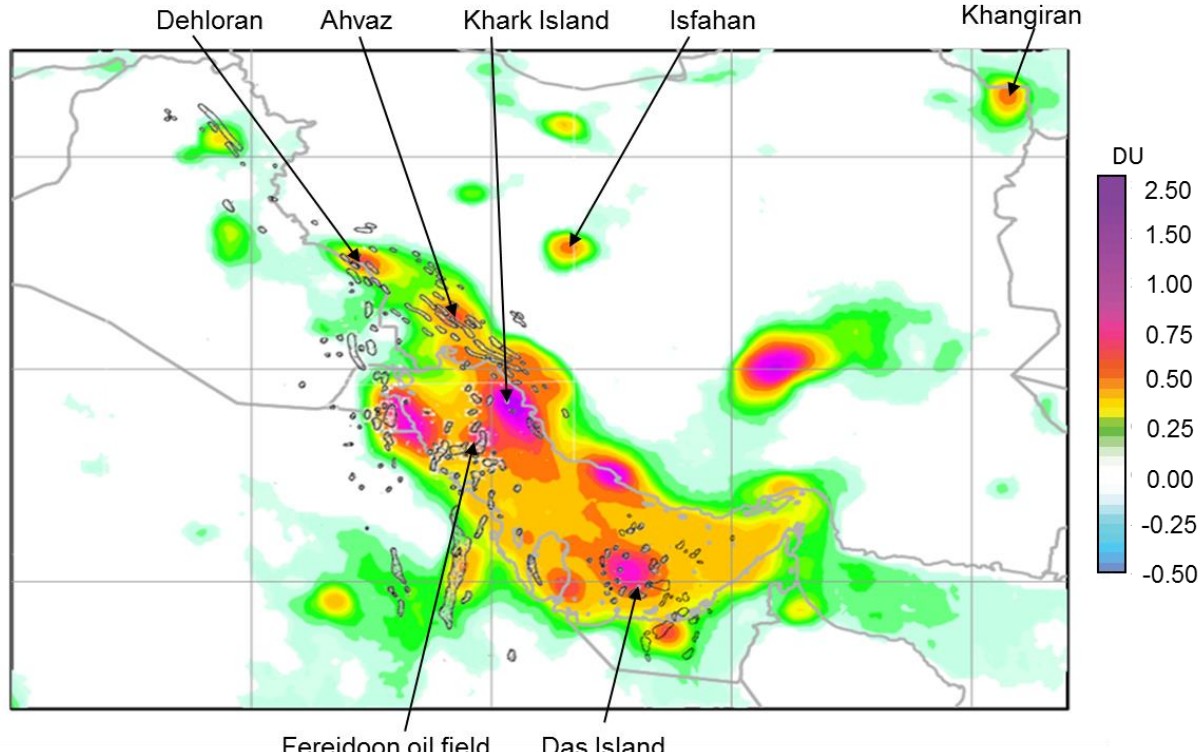

**Figure 8.** Mean SO$_2$ distribution over the Persian Gulf in 2005-2007. The black contours indicate the main oil fields. Examples of SO$_2$ sources related to the oil and gas industry are also shown: the Khark Island terminal and refinery, Isfahan oil refinery and power plant, Khangiran gas refinery, Das Island oil rigs as well as oil and gas storage and terminal, and oil rigs in the Fereidoon oil field. Sources Dehloran and Ahvaz are located near major oil fields and are possibly related to flaring.





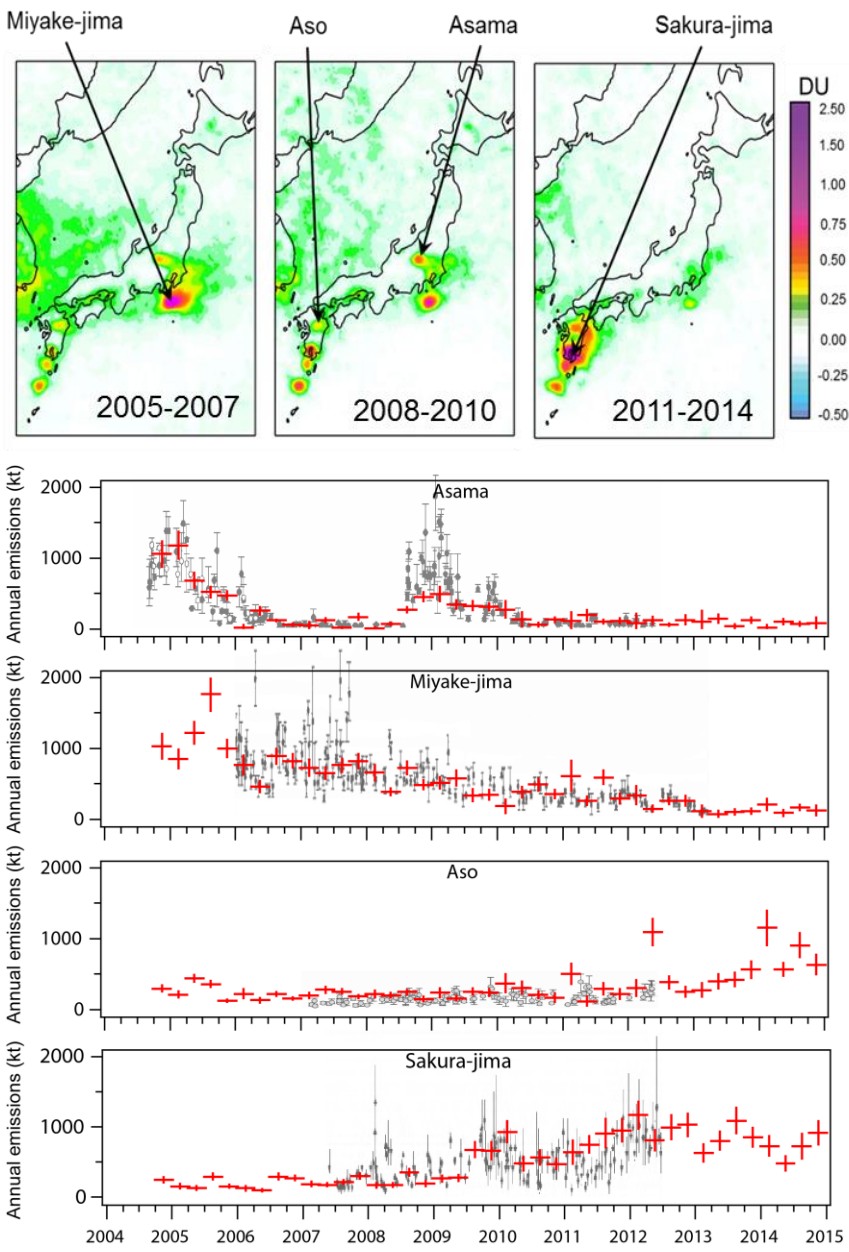

**Figure 9.** (top) Maps of the mean SO$_2$ values for 2005-2007 and 2008-2010 over Japan. (bottom) Time series of OMI-
estimated annual emission rates calculated for every 3 months for four volcanoes are show in red. The error bars represent
2-σ confidence intervals. Grey dots are daily emission estimates provided by the Japan Meteorological Agency. The grey
vertical bars represent minimum and maximum values.

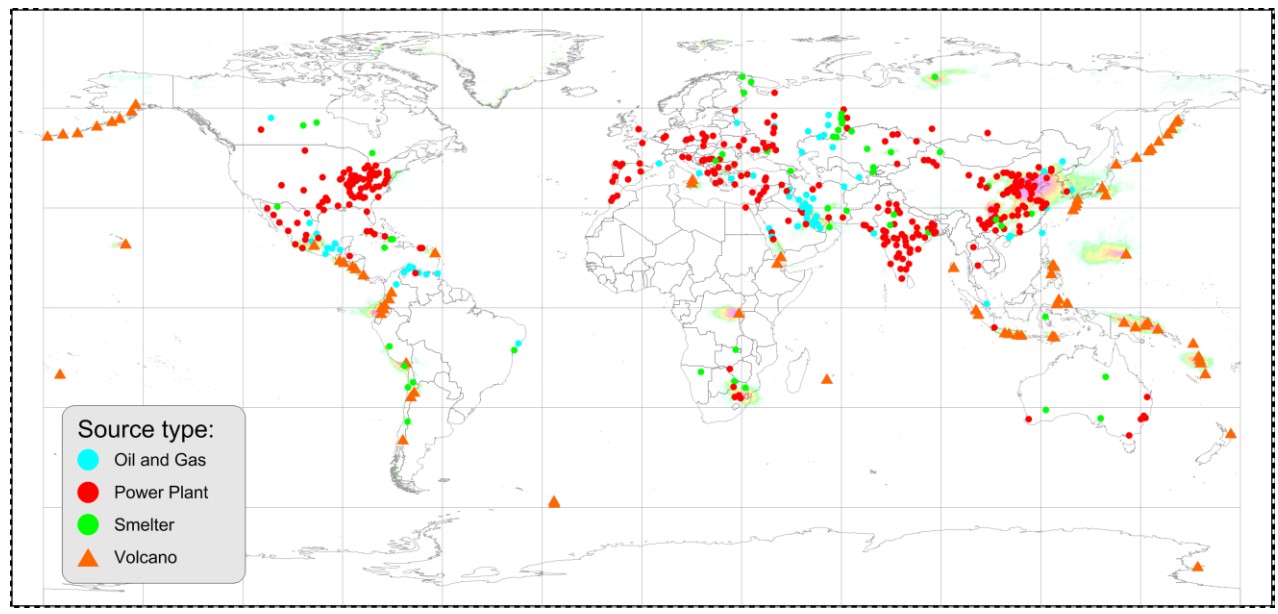

**Figure 10.** Geographic distribution of the OMI-based SO₂ sources in the catalogue. At present, there are 491 sites based on 2005-2007 data (297 Power Plants, 53 Smelters, 65 Oil and Gas industry-relates sources, and 76 Volcanoes).

**Figure 11.** (top) Strength of SO$_2$ emission sources in (top) 2005-2007 and (middle) 2012-2014. The size of the symbols is proportional to the mean annual emission values. (bottom) Relative changes between 2005-2007 and 2012-2014 emission values as percentage of the mean 2005-2014 annual emissions. Only sources with mean 2005-2014 emissions greater than 30 kt are shown. Note that some dots in China represent area sources.





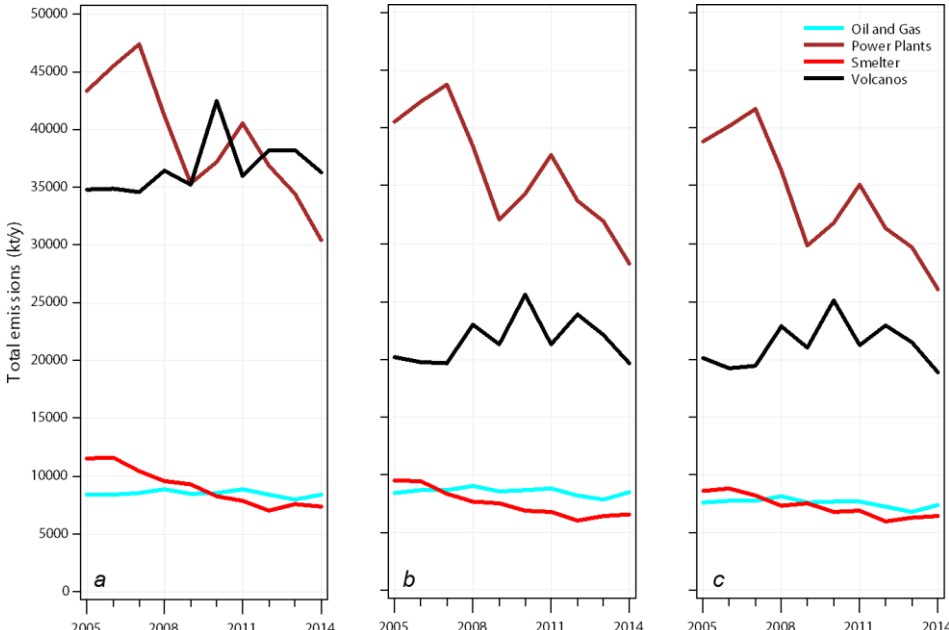

**Figure 12**. Total annual emissions by source type: power plants, smelters, sources related to the oil and gas industry, and volcanic sources. Emissions were calculated using constant AMF=0.36 (a), an AMF value that depends on the site altitude only (b), and an AMF value that was calculated using the site altitude, albedo, and the PBL height (c).




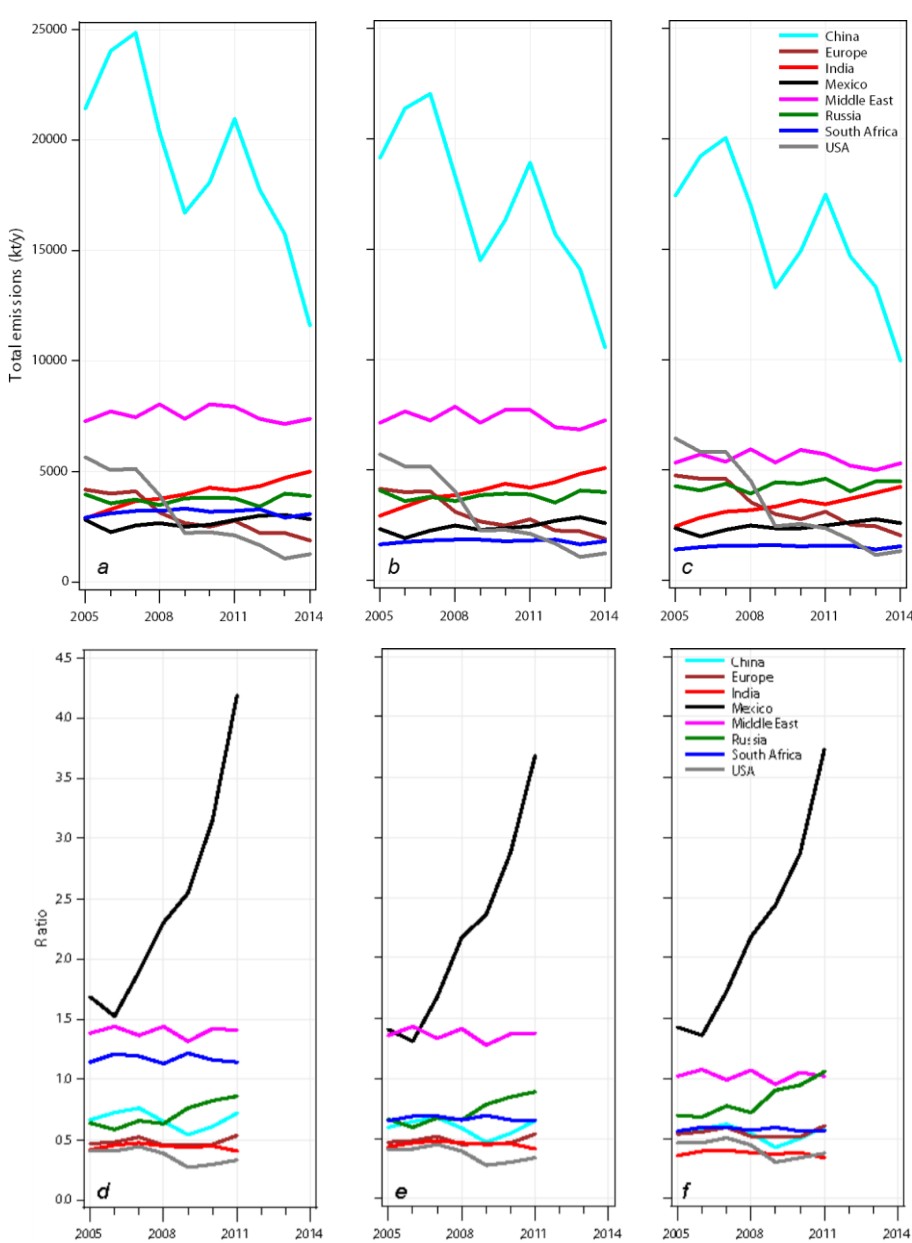

**Figure 13**. (a-c) Time series of total annual $SO_2$ emissions by country/region calculated for the period 2005-2014, and (d-f) time series of the ratios between OMI-estimated and reported annual emissions by country/region. Emissions were calculated using constant AMF=0.36 (a and d), AMF values that depend on the site altitude only (b and d), and AMF values that were calculated using the site altitude, albedo, and PBL height (c and f). Note that western and central Europe are labeled jointly as "Europe".