# Peer review of "A global catalogue of large SO2 sources and emissions derived from the Ozone Monitoring Instrument"

_Atmospheric Chemistry and Physics, 2016_

## Referee Comment (RC1) · A. J. Krueger (Referee) · 20 Jun 2016

A uniform census of large global SO2 sources is a significant advancement in monitoring the inputs to the atmosphere for air quality and climate change assessment. This well written paper makes use of data from the second generation of space-borne UV mapping spectrometers to detect and quantify global ground level sources of sulfur dioxide. While large volcanic eruption SO2 cloud masses have been measured for over 35 years starting with discrete wavelength TOMS instruments, detection of industrial sources in the boundary layer need the full spectrum of backscattered sunlight available through new technology in the OMI and GOME instruments, among others. This paper also takes advantage of the OMI Principle Component SO2 retrieval algorithm

for PBL sources with its doubled sensitivity and reduced regional biases. New source detection and analysis tools from Fioletov et al., 2015 are then used in calculation of source strengths. The wind rotation and upwind - downwind difference methods are clever ways to locate and measure sources for inclusion in the census.

The scope of the paper is broad and a challenge to review in depth. My review focuses on technical details rather than environmental and industrial issues. However the significance of this dataset for identifying "missing" sources is great and should lead to more accountability in the bottom-up emissions estimates. For example, it's easy to see in the Appendix that the Ilo smelter in Peru reduced its high emissions down to zero by 2009 and kept it there through 2013, but emissions returned again in 2014 almost to 2007 levels. The failures of bottom up emission inventories, such as in Mexico for whatever reason, will become obvious.

The error analysis shown in Table 1 seems to be conservative but the large size of some error sources like the 35% uncertainty from lifetime and width assumptions suggests that these errors could be reduced through refinement of SO2 decay rates at individual sites.

This work can resolve questions about the relative contributions of volcanoes and anthropogenic sources. A global inventory of SO2 from degassing volcanoes is significant contribution.

This research study is another success for the NASA MEaSUREs program. I recommend publication after relatively minor revisions.

General Comments

This paper depends on information in Fioletov et al., 2015 to be understood. For example, "Fitting Uncertainty" is not defined except as a standard deviation of the estimated emissions. It would be good to point the reader to the 2015 paper in the Introduction. A brief summary of the techniques presented in that paper would resolve questions that

arise in Section 3 and the Appendix.

P 5, Section 3.1 A discussion of sample sizes is missing. How many samples are needed to detect a source? How many samples are collected to compute the annual emission value? What is the time resolution in trends?

P 7. It is a concern that an average lifetime of 6 hours is used for all sources although it must depend on variable factors, like humidity. Is it practical to fine-tune the lifetime if meteorological data are available?

Figure 12 shows a time series of total annual $SO_2$ emissions for the four primary source types while Figure 13 shows the total $SO_2$ emissions for separate countries or regions and the ratio to reported emissions. Both figures show the results for three different AMF choices. The main points of the figures are lost when AMF information is included. It has already been made clear that the fixed value of 0.36 in the production dataset does not apply globally or to all individual stations. Unless the AMF choice influences the time series why include it here redundantly? If the 3rd case is the best why not just show that and perhaps condense the effects of AMF choice into a few sentences or a summary chart. Then the real info about the trends and ratio of reported to satellite annual emissions will become the focus of the figures.

The catalogue in the Appendix contains a single value for the AMF for each site. Is this the average of the AMF's for each of the observations at that site or the result of prescribed conditions? If so how are the effects of changing conditions on the AMF accounted for?

The range of AMF's in the source dataset is unexpectedly large (< 0.3 - > 1.8). I can understand the value of 1.87 for Erebus volcano due to permanent snow/ice cover and altitude. But the substantial variability implies that the PCA dataset is not representative of global conditions. However, it is probably unreasonable to compute global values of AMF for each OMI pixel. It seems that the current approach of computing AMF's just the source locations is probably the best.

One piece of information missed in the Appendix is the sample size for each source. How many observations go into each annual emission data point? You might add a new page in the workbook to include this information.

Specific Comments:

P 6, L 18ff. The choice of the letter, a, to represent SO2 mass is awkward and leads to confusion with the article, a (P 7, L 6). I realize that a was used previously for this variable but I would consider changing to a less ambiguous symbol.

P 7, L 6: The phrase "depends on a linearly." should be "depends linearly on a." This would not be as clumsy if "a" were replaced by "m" or some other letter or symbol.

P 10, L 11. If the use of SO2 cross sections for a stratospheric temperature (203K) in DOAS boundary layer retrievals creates a 19 % difference from PCA (283K) retrieved values how can the emission estimates be the same within 5%. Please elaborate on how the emission algorithm could decrease the bias.

P 11. The term "emission-factor" is not defined. Please correct this.

P 16, Sec. 5.4 Volcanoes: In the context of this paper on the detection and evaluation of SO2 sources in the PBL I think the limitations of measuring volcanic emissions is overstated. McCormick et al and the other cited authors have rightfully addressed the possible sources of error in volcanic plume retrievals in their papers. However, the majority of these plumes are released into the free troposphere above the PBL where the AMF is readily computed. In addition any ash falls out rapidly near the volcano. The "numerous issues such as limited instrument sensitivity to volcanic plumes at low altitudes" seem simple when compared with the large uncertainties in the measurement of industrial emissions in the PBL.

Snow covered volcanic cones might affect a single OMI pixel and the plume drifts free of the volcano peak over the lower altitude terrain with its seasonal changes of reflectivity. Thus I don't believe "Albedo effects from snow-covered volcanic cones" is a significant

problem (except for Erebus, where constant snow cover seems to be accounted for in the Catalogue). For passively degassing volcanoes, the vent height is generally known and the SO2 plume is usually released above the PBL, so detection is better than industrial sources. The NASA OMI BRD dataset provides solutions for different altitude regimes to allow the user to choose the appropriate altitude for the plume. The PCA dataset as implemented by GSFC focused on air quality and PBL emissions so the data need to be corrected for other altitude regimes, as the authors have done.

P 19, L 30. The phrase 'sand covered areas with high albedos" seems misleading. The UV albedo of sand ($\sim$10%) is not large relative to the visible wavelengths although it is larger than for vegetation.

Figure 2. The term "Nobs" is not in my vocabulary. Can you define it in the text or replace it with a common term?

Technical corrections:

P 4,L 7. The exponent in 10ˆ26 is mislocated.

P 6, L 4 and P 7, L 18. Typo; "centered" instead of "cantered".

P 13, L 5. Use "ground" instead of "grounded".

P 31, Table 1. Note for Lifetime and Width: delete "Used" in "Used prescribed. . ..."

P 35, Figure 3. Subplot identifier, d, has escaped to the first line of the caption. All the identifiers could be made more prominent.

Throughout the text, in-line references are fully enclosed in parentheses. e.g. on p 11, 'the value reported by (Kaldellis et al., 2004)'. In other journals this would be 'the value reported by Kaldellis et al.,(2004)'.

Figure 13 caption: line 5. (b and e), not (b and d)

References:

[Figure]

The Beirle et al paper is referenced twice; both the discussion and the published paper are included. I assume the latter replaces the discussion paper and the former should be deleted.

Attachment:

ReadMe page;

Line 4 Col B: "Longitude" misspelled

Line 6, Col B: "Source" misspelled.

---

## Referee Comment (RC2) · Anonymous Referee #2 · 25 Aug 2016

Review of the ACPD paper by Fioletov et al., 2016

The authors present a very impressive global catalogue of point sources of SO2 with additional annual emission estimates, based on a well-documented methodology that uses OMI columnar SO2 data. Concerning the methodology the authors present additionally in this paper uncertainty estimates relevant to the choice of the satellite algorithm, the AMF calculation, the estimation of mass and the fitting procedures. They conclude that their methodology is sensitive to sources stronger than 30kt/yr with an overall uncertainty close to 50%. The paper should be accepted for publication almost as it is, considering the correction of few minor typos and some comments below:

Page 3, line 26. The authors use the period 2005-2007 for the identification of the

[Figure]

sources. What about sources that could appear after this period? Can this be excluded? Please comment on that.

Page 4, line 6. Check the equation. There is no exponent.

Section 2. After reading section 2 the reader is confused concerning the differences in AMF estimations between the two algorithms and their consistency. Please provide some more information for the BIRA algorithm on that.

Section 7. In this section it could be good to provide a comment on the uncertainty of the bottom-up inventories and compare this to the uncertainty of the retrieved emissions.

---

## Author Comment (AC1) · 30 Aug 2016

A. J. Krueger (Referee)

A uniform census of large global SO2 sources is a significant advancement in monitoring the inputs to the atmosphere for air quality and climate change assessment. This well written paper makes use of data from the second generation of space-borne UV mapping spectrometers to detect and quantify global ground level sources of sulfur dioxide. While large volcanic eruption SO2 cloud masses have been measured for over 35 years starting with discrete wavelength TOMS instruments, detection of industrial sources in the boundary layer need the full spectrum of backscattered sunlight available through new technology in the OMI and GOME instruments, among others. This paper also takes advantage of the OMI Principle Component SO2 retrieval algorithm for PBL sources with its doubled sensitivity and reduced regional biases. New source detection and analysis tools from Fioletov et al., 2015 are then used in calculation of source strengths. The wind rotation and upwind - downwind difference methods are clever ways to locate and measure sources for inclusion in the census.

The scope of the paper is broad and a challenge to review in depth. My review focuses on technical details rather than environmental and industrial issues. However the significance of this dataset for identifying "missing" sources is great and should lead to more accountability in the bottom-up emissions estimates. For example, it's easy to see in the Appendix that the Ilo smelter in Peru reduced its high emissions down to zero by 2009 and kept it there through 2013, but emissions returned again in 2014 almost to 2007 levels. The failures of bottom up emission inventories, such as in Mexico for whatever reason, will become obvious.

The error analysis shown in Table 1 seems to be conservative but the large size of some error sources like the 35% uncertainty from lifetime and width assumptions suggests that these errors could be reduced through refinement of SO2 decay rates at individual sites.

This work can resolve questions about the relative contributions of volcanoes and anthropogenic sources. A global inventory of SO2 from degassing volcanoes is significant contribution.

This research study is another success for the NASA MEaSUREs program. I recommend publication after relatively minor revisions.

We would like to thank Dr. Arlin Krueger his very thorough revive and detailed comments that helped us improve the manuscript.

General Comments

This paper depends on information in Fioletov et al., 2015 to be understood. For example, "Fitting Uncertainty" is not defined except as a standard deviation of the estimated emissions. It would be good to point the reader to the 2015 paper in the Introduction. A brief summary of the techniques presented in that paper would resolve questions that arise in Section 3 and the Appendix.

We pointed to the Fioletov et al., 2015 paper in the introduction. We also changed the column title in the Supplement to "UNCERTAINTY (one standard deviation of the estimated emissions)" to make it clear.

P 5, Section 3.1 A discussion of sample sizes is missing. How many samples are needed to detect a source? How many samples are collected to compute the annual emission value? What is the time resolution in trends?

Emissions in the paper are estimated from a linear regression model determined by just one parameter, so in theory, just one pixel could be enough for such estimate. In reality, of course, data from many orbits are needed to get an emission estimate. Also, it is not just a number of pixels, but also their location relative to the source (upwind or downwind) affects the estimate uncertainty. In the original manuscript, we had already calculated uncertainties for every annual emission estimate to give information on the significance of the estimated value.

It may be possible to calculate seasonal and even monthly emissions, but then seasonal changes in observational and weather conditions would start to play a major role. For this reason we focused this study on annual emissions. We added a sentence about this to the Summary.  To address reviewer's questions about the sample size, we added a table to the supplement with the number of pixels within the fitting domain that were used to calculate the annual emission values.

P 7. It is a concern that an average lifetime of 6 hours is used for all sources although it must depend on variable factors, like humidity. Is it practical to fine-tune the lifetime if meteorological data are available?

6 hours was an average lifetime used to calculate annual emissions.  The sensitivity analysis in this study shows that this value works reasonably well for such estimates. The lifetime could be different in different seasons, it would also depend on meteorological conditions. But it is really outside the scope of this study. As mentioned above, we added a sentence about seasonal and weather conditions to the Summary. Note also that the impact of using a constant lifetime was incorporated into the uncertainty budget.

Figure 12 shows a time series of total annual SO2 emissions for the four primary source types while Figure13 shows the total SO2 emissions for separate countries or regions and the ratio to reported emissions. Both figures show the results for three different

AMF choices. The main points of the figures are lost when AMF information is included. It has already been made clear that the fixed value of 0.36 in the production dataset does not apply globally or to all individual stations. Unless the AMF choice influences the time series why include it here redundantly? If the 3rd case is the best why not just show that and perhaps condense the effects of AMF choice into a few sentences or a summary chart. Then the real info about the trends and ratio of reported to satellite annual emissions will become the focus of the figures.

The trends and ratio have been discussed in another paper published in Nature Geoscience (McLinden et al., 2016), but the AMF effects and other technical information were omitted in that paper due to the size limitation. We believe that it is important to highlight the significance of AMF information on the final results. This is why we included Figures 12 and 13. We changed the text in the first paragraph of Section 7 to make this clear.

The catalogue in the Appendix contains a single value for the AMF for each site. Is this the average of the AMF's for each of the observations at that site or the result of prescribed conditions? If so how are the effects of changing conditions on the AMF accounted for?

We did not calculate AMF for each observation. Instead, it was estimated for each site by a radiative transfer model based on the site elevation, representative viewing geometry, surface albedo, and other parameters. It is discussed in the second paragraph of Section 2. We added a sentence "As a result, a single site-specific AMF value for each site was calculated" to that paragraph to avoid confusion.

The range of AMF's in the source dataset is unexpectedly large (< 0.3 - > 1.8). I can understand the value of 1.87 for Erebus volcano due to permanent snow/ice cover and altitude. But the substantial variability implies that the PCA dataset is not representative of global conditions. However, it is probably unreasonable to compute global values of AMF for each OMI pixel. It seems that the current approach of computing AMF's just the source locations is probably the best.

Yes, this is exactly how it was done in the paper. A single site-specific AMF value was calculated for each catalogue site.

One piece of information missed in the Appendix is the sample size for each source. How many observations go into each annual emission data point? You might add a new page in the workbook to include this information.

We added a table to the Supplement that contains the number of pixels used to calculate each annual emission value for each site. Note that we used 3 different domain areas to do the calculations: the larger the source, the bigger the domain area. In order to make the sample size result consistent, we reported the number of pixels in the smallest domain area.

Specific Comments:

P 6, L 18ff. The choice of the letter, a, to represent SO2 mass is awkward and leads to confusion with the article, a (P 7, L 6). I realize that a was used previously for this variable but I would consider changing to a less ambiguous symbol.

We replaced "a" with alpha ($\alpha$) to avoid this confusion. They appear different but the formulas look more or less the same as in the earlier papers.

P 7, L 6: The phrase "depends on a linearly." should be "depends linearly on a." This would not be as clumsy if "a" were replaced by "m" or some other letter or symbol.

Corrected with "a" replaced by alpha

P 10, L 11. If the use of SO2 cross sections for a stratospheric temperature (203K) in DOAS boundary layer retrievals creates a 19 % difference from PCA (283K) retrieved values how can the emission estimates be the same within 5%. Please elaborate on how the emission algorithm could decrease the bias.

We simply adjusted BIRA data for the difference in AMF and cross sections. We added a sentence " We adjusted BIRA DOAS data for this temperature effect and for the difference in the AMF factors to match NASA PCA data for the comparison" to make this clear.

P 11. The term "emission-factor" is not defined. Please correct this.

We added the explanation (the amount of released SO2 per Megawatt)

P 16, Sec. 5.4 Volcanoes: In the context of this paper on the detection and evaluation of SO2 sources in the PBL I think the limitations of measuring volcanic emissions is overstated. McCormick et al and the other cited authors have rightfully addressed the possible sources of error in volcanic plume retrievals in their papers. However, the majority of these plumes are released into the free troposphere above the PBL where the AMF is readily computed. In addition any ash falls out rapidly near the volcano. The "numerous issues such as limited instrument sensitivity to volcanic plumes at low altitudes" seem simple when compared with the large uncertainties in the measurement of industrial emissions in the PBL.

Snow covered volcanic cones might affect a single OMI pixel and the plume drifts free of the volcano peak over the lower altitude terrain with its seasonal changes of reflectivity. Thus I don't believe "Albedo effects from snow-covered volcanic cones" is a significant problem (except for Erebus, where constant snow cover seems to be accounted for in the Catalogue). For passively degassing volcanoes, the vent height is generally known and the SO2 plume is usually released above the PBL, so detection is better than industrial sources. The NASA OMI BRD dataset provides solutions for

different altitude regimes to allow the user to choose the appropriate altitude for the plume. The PCA dataset as implemented by GSFC focused on air quality and PBL emissions so the data need to be corrected for other altitude regimes, as the authors have done.

The present PCA data set was not really designed to deal with volcanic SO2 and we wanted to warn a reader about this. But perhaps we overstated the problems, the main issue, the plume height was handled by establishing site-specific AMFs. We changed the tone of this section be replacing "has numerous issues" with "may be affected by". We also added that "… we have corrected PCA data using altitude-dependent AMFs as described in section 2 that largely removed altitude-related biases of the standard PCA data set that were the main source of errors in the volcanic SO2 estimates.." to make it clearer.

P 19, L 30. The phrase 'sand covered areas with high albedos" seems misleading. The UV albedo of sand (~10%) is not large relative to the visible wavelengths although it is larger than for vegetation.

We changed it to "where the albedo is higher than over water or vegetation "

Figure 2. The term "Nobs" is not in my vocabulary. Can you define it in the text or replace it with a common term?

We added a sentence that defines "Nobs"

Technical corrections:

P 4,L 7. The exponent in 10^26 is mislocated.

Corrected

P 6, L 4 and P 7, L 18. Typo; "centered" instead of "cantered".

Corrected

P 13, L 5. Use "ground" instead of "grounded".

Corrected

P 31, Table 1. Note for Lifetime and Width: delete "Used" in "Used prescribed . ."
Corrected

P 35, Figure 3. Subplot identifier, d, has escaped to the first line of the caption. All the identifiers could be made more prominent.

Corrected

Throughout the text, in-line references are fully enclosed in parentheses. e.g. on p 11, 'the value reported by (Kaldellis et al., 2004)'. In other journals this would be 'the value reported by Kaldellis et al.,(2004)'.

This was caused by the reference managing software and this was corrected where necessary.

Figure 13 caption: line 5. (b and e), not (b and d)

Corrected

The Beirle et al paper is referenced twice; both the discussion and the published paper are included. I assume the latter replaces the discussion paper and the former should be deleted.

Corrected as suggested

Attachment:

ReadMe page;
Line 4 Col B: "Longitude" misspelled
Line 6, Col B: "Source" misspelled.

Both corrected.

---

## Author Comment (AC2) · 30 Aug 2016

The authors present a very impressive global catalogue of point sources of SO2 with additional annual emission estimates, based on a well-documented methodology that uses OMI columnar SO2 data. Concerning the methodology the authors present additionally in this paper uncertainty estimates relevant to the choice of the satellite algorithm, the AMF calculation, the estimation of mass and the fitting procedures. They conclude that their methodology is sensitive to sources stronger than 30 kt/yr with an overall uncertainty close to 50%. The paper should be accepted for publication almost as it is, considering the correction of few minor typos and some comments below:

We would like to thank the reviewer for the evaluation and comments that helped us improve the manuscript.

Page 3, line 26. The authors use the period 2005-2007 for the identification of the sources. What about sources that could appear after this period? Can this be excluded? Please comment on that.

The 2005-2007 period was used as the main period to detect the sources. We performed a similar check for the 2008-2010 and 2011-2014 periods, but not as thoroughly as for the 2005-2007 since these data were affected by the row anomaly after 2006. As a result, the noise was increased and the criteria used for the 2005-2007 data could not be applied any longer.  The following sentence was added: "We also looked at the 2008-2010 and 2011-2014 periods and identified sources that appeared at that time, although the source detection limit began to deteriorate after 2006 due to the row anomaly and its continued expansion (see e.g., (Krotkov et al., 2016) for details) "

Page 4, line 6. Check the equation. There is no exponent.

Corrected

Section 2. After reading section 2 the reader is confused concerning the differences in AMF estimations between the two algorithms and their consistency. Please provide some more information for the BIRA algorithm on that.

We agree that the text creates some confusion, although more details are given later in Section 4. To help clarify, we added a sentence " However, for the purpose of comparison between OMI PCA and BIRA DOAS algorithms in this study, we converted slant columns into vertical columns using a constant AMF".

Section 7. In this section it could be good to provide a comment on the uncertainty of the bottom-up inventories and compare this to the uncertainty of the retrieved emissions

We added a sentence about the bottom-up inventory uncertainties; "It should be mentioned that the bottom-up inventories also have some uncertainties. Smith et al., (2011) estimated the uncertainty bounds (as 95% confidence interval) in ±11%, ±21%, and ±14% for the coal, oil and gas and smelting industries, respectively."